# Exploiting Negative Samples: A Catalyst for Cohort Discovery in Healthcare Analytics

## Abstract

In healthcare analytics, particularly when dealing with binary diagnosis or prognosis tasks, unique challenges arise from the inherent asymmetry between positive and negative samples. Positive samples, denoting patients who develop a disease, are defined based on stringent medical criteria. In contrast, negative samples are defined in an open-ended manner, leading to a vast potential set. Despite this fundamental asymmetry, the role of negative samples remains underexplored in prior research, possibly due to the enormous challenge of investigating an infinitely large negative sample space. To bridge this gap, we propose an innovative approach to facilitate cohort discovery within negative samples, leveraging a Shapley-based exploration of interrelationships between these samples, which holds promise for uncovering valuable insights concerning the studied disease, and related comorbidity and complications. We quantify each sample's contribution using data Shapley values, subsequently constructing the Negative Sample Shapley Field to model the distribution of all negative samples. Next, we transform this field through manifold learning, preserving the essential data structure information while imposing an isotropy constraint in data Shapley values. Within this transformed space, we pinpoint cohorts of medical interest via density-based clustering. We empirically evaluate the effectiveness of our approach on our hospital's electronic medical records. The medical insights derived from the discovered cohorts are validated by clinicians, which affirms the medical value of our proposal in unveiling meaningful insights aligning with existing domain knowledge, thereby bolstering medical research and well-informed clinical decision-making.

## 1 Introduction

Healthcare analytics leverages diverse healthcare data sources to perform many analytic tasks including diagnosis (Lipton et al., 2016) and prognosis (Mould, 2012). Electronic Medical Records (EMR) are perhaps the most important of these data sources, since they play a crucial role in recording patients' essential information and providing a comprehensive view of their health conditions. The recently increasing availability of EMR data has spawned the development of healthcare analytics models for effective patient management and medical resource allocation.

Without loss of generality, let us delve into a diagnosis or prognosis problem of predicting whether a patient has developed/will develop a certain disease based on the EMR data. This problem is a binary classification, where patients who develop the disease are "positive samples", while those who do not are "negative samples". Notably, we identify the unique nature of such binary classifications in healthcare analytics, as compared to traditional classification tasks. For instance, when classifying cats vs. dogs, both positive and negative samples are based on objective facts. However, in healthcare analytics, positive samples are defined according to rigorous medical criteria, based on medical theories and experience. Contrarily, negative samples are defined in an unrestricted manner, as the complementary set of the positive samples. Consequently, the set of negative samples may encompass a vast number of diverse individuals who are outside the scope of the studied disease or who are healthy. This leads to an inherent asymmetry between positive and negative samples, as positive samples are well-defined and bounded, while negative samples are diverse and open-ended.

Despite such fundamental asymmetry in healthcare analytics, previous research has not adequately addressed the role of negative samples. One potential reason for this research gap is the enormous challenge posed by investigating an infinitely large negative sample space, which cannot be easily addressed using existing approaches, e.g., it could be difficult to understand why general healthy

individuals do not develop a disease. Nonetheless, it is crucial to probe into negative samples for a more comprehensive investigation of the studied disease. Although it may not have developed in these samples, some may exhibit similar symptoms or even develop related conditions such as its comorbidity or complications. Hence, these negative samples are in urgent need of close medical attention, as they provide an opportunity for clinicians to gain a deeper understanding of the studied disease, leading to more accurate and comprehensive diagnoses, prognoses, and treatment plans.

In this paper, we aim to address this gap by exploring negative samples in healthcare analytics. Given the diversity of negative samples, it may not be meaningful to consider them all as one "group". Instead, we examine the underlying distribution of negative samples to automatically identify medically insightful groups of patients with shared characteristics, referred to as "cohorts" (Mahmood et al., 2014; Zhou et al., 2020). Such cohort discovery among negative samples can provide fresh insights to clinicians on the studied disease, e.g., comprehending the factors contributing to the absence of the disease and the development of related conditions.

**Solution.** As front-line clinicians and medical researchers, we bring a unique perspective to guide our methodology design in effectively discovering cohorts among negative samples. In Sec. 3, we elaborate on our approach with three components. Firstly, we propose to quantify each negative sample's contribution to the prediction task using data Shapley values (Rozemberczki et al., 2022; Ghorbani & Zou, 2019). We then construct the Negative Sample Shapley Field, an inherently existing scalar field describing the distribution of all negative samples (Sec. 3.1). Secondly, to effectively discover cohorts, we transform the original field by manifold learning (Bengio et al., 2013) while preserving the original data structure information and ensuring that changes in data Shapley values are isotropic in all orientations (Sec. 3.2). Thirdly, in the transformed manifold space, we identify densely-connected clusters among the negative samples with high data Shapley values through DBSCAN (Sec. 3.3). These clusters help us locate "hot zones", which are our desired cohorts to discover, exhibiting similar medical characteristics with high data Shapley values.

**Novelty.** (i) In contrast to mainstream medical cohort studies, we adopt a distinct perspective by focusing on negative samples, and emphasize the significance of cohort discovery among negative samples, as they can reveal future positives, pathological correlations, or similar conditions. This reciprocal relationship between negative and positive samples could contribute to defining positive samples in theoretical medical research. (ii) Existing studies on data Shapley values predominantly measure the value of individual data samples, e.g., for federated learning, or apply them at finer levels for feature explainability (Rozemberczki et al., 2022). What distinguishes our paper is its innovative extension of this concept to a Shapley-based exploration of interrelationships between samples. This extension goes beyond traditional feature-based similarity methods, asserting that valuable cohorts should exhibit similar distributions with high data Shapley values.

**Contributions.** (i) We bridge the research gap caused by the asymmetry between positive and negative samples in healthcare analytics by exploring negative samples for cohort discovery. (ii) We propose an innovative approach for effective cohort discovery: constructing the Negative Sample Shapley Field, transforming the field by manifold learning with structure preservation and isotropy constraint, and discovering cohorts in the manifold space via DBSCAN. (iii) We empirically evaluate the effectiveness of our approach using our hospital's EMR (Sec. 4). Our approach unveils meaningful insights that align with existing domain knowledge and have been verified by clinicians. These findings have the potential to benefit medical practitioners by facilitating medical research and enhancing clinical decision-making in healthcare delivery.

## 2 PROBLEM AND OUR SOLUTION

**Distinctiveness of negative samples and the unbounded negative sample space.** Let us take hospital-acquired acute kidney injury (AKI), a disease we strive to handle in practice, as an example. In this AKI prediction task, we aim to predict if a patient will develop AKI in the future. A positive sample is a patient who meets the stringent KDIGO criteria (Kellum et al., 2012), and has a closed definition, whereas a negative sample has an open definition without restrictions. Hence, negative samples form an unbounded space, demonstrating an asymmetry compared to positive samples.

**Construction of the Negative Sample Shapley Field for cohort discovery.** To facilitate the analysis of negative samples, we need to investigate their distribution and identify those that are most relevant to the prediction task (e.g., AKI prediction task above) and hence worth exploring. In this regard, we propose to measure the valuation of each negative sample to the task by its data Shapley

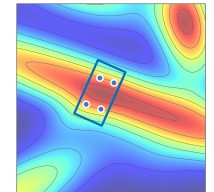 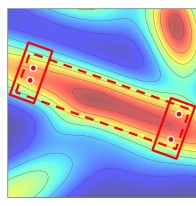 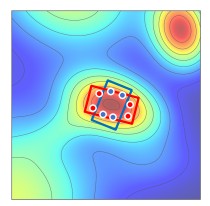

(a) Discovered hot zone in the Negative Sample Shapley Field by clustering high-value negative samples

(b) Mis-discovered hot zones in the Negative Sample Shapley Field

(c) Manifold space integrating data structure information and isotropy constraint

Figure 1: Discovery of hot zones in the Negative Sample Shapley Field.

value. Based on such valuations, we construct a scalar field, the Negative Sample Shapley Field, in which each point is a negative sample, and the point's value is its data Shapley value. This field depicts the distribution of negative samples (see Figure 1(a) for an example). We define **"hot zones"** in this field, identified by points with high data Shapley values, as **"cohorts"**. Our objective is to automatically detect these cohorts, revealing medically meaningful patterns.

**Cohort discovery via manifold learning and density-based clustering.** We note that the vast number of negative samples renders an exhaustive search infeasible. Although the Negative Sample Shapley Field is continuously differentiable, the high computational overhead makes it intractable to find local optima via gradient descent. To overcome this obstacle, we make the assumption that a subset of negative samples collected in clinical practice carries significant medical value, e.g., patients who visit hospitals for examinations but do not develop the disease. We posit that these real-world negative samples should be proximate to our desired hot zones in the space and can effectively sample our hot zone boundaries, which are hence of medical interest.

In Figure 1, we exemplify how to discover hot zones in the Negative Sample Shapley Field. Figure 1(a) and (b) demonstrate four points situated on the same contour line, indicating their inclusion in the same hot zone. However, only the former case yields the expected discovered cohort, while the latter leads to mis-discovery. This highlights that the originally constructed Negative Sample Shapley Field is suboptimal for cohort discovery among negative samples, due to its anisotropy in data Shapley values. To overcome this issue, we propose a manifold learning approach. Specifically, we leverage manifold learning to reduce the dimensionality of the raw, sparse EMR data to derive compact representations that not only preserve the underlying data structure information but also benefit subsequent spatial clustering analysis. Further, we introduce an isotropy constraint to ensure uniform changes in data Shapley values across all orientations, which prevents the mis-discovery as in Figure 1(b). This transformed space, integrating the data structure information and the isotropy constraint, is more suitable for subsequent cohort discovery as illustrated in Figure 1(c).

Our objective is then to identify medically meaningful cohorts, specifically dense regions formed by negative samples with high data Shapley values in the manifold space. We set a data Shapley value threshold to extract negative samples with high values and employ the DBSCAN algorithm to detect the hot zones among them. The derived cohorts could shed light on the studied disease, its related comorbidity, and complications, thereby empowering clinicians in practical healthcare delivery.

## 3 METHODOLOGY

### 3.1 NEGATIVE SAMPLE SHAPLEY FIELD CONSTRUCTION

Given EMR data $\mathcal{D} = \{d_i\}$, where $d_i$ is a sample with $i \in \{0, \ldots, N-1\}$ and $N$ denotes the total sample number. We focus on binary classification, and each $d_i$ consists of input features and a binary label. To investigate negative samples for cohort discovery, we divide $\mathcal{D}$ into $\mathcal{D}^+$ and $\mathcal{D}^-$, representing positive and negative samples. We denote $\mathcal{D}^- = \{d_i^-\}$, where $d_i^-$ is a negative sample with $i \in \{0, \ldots, N^- - 1\}$ and $N^-$ is the negative sample number.

Each negative sample $d_i^- = (\mathbf{x}_i, y_i)$ comprises the input features $\mathbf{x}_i$ and its binary label $y_i$. Our objective is to measure the value of each negative sample by quantifying its contribution to the prediction performance, which we refer to as data valuation. The data Shapley value (Ghorbani & Zou, 2019), stemming from the Shapley value in cooperative game theory, has made significant advances in data valuation (Rozemberczki et al., 2022), which inspires our proposal to calculate the data Shapley value of each negative sample as its value. Specifically, let $F$ denote the prediction model and suppose we are interested in evaluating $F$'s performance on a subset of negative samples

$\mathcal{Q} \subseteq \mathcal{D}^-$, along with all the positive samples $\mathcal{D}^+$. We define $M$ as the performance metric function, and then $M(\mathcal{D}^+ \cup \mathcal{Q}, F)$ is the performance achieved on the combined set of $\mathcal{D}^+$ and $\mathcal{Q}$. We define $s_i$ as the data Shapley value for the negative sample $d_i^-$. $s_i$ satisfies three properties of Shapley values: (i) null player, (ii) symmetry, and (iii) linearity, which are the essential properties of an equitable data valuation (Ghorbani & Zou, 2019). We calculate $s_i$ as follows.

**Proposition 1** *The data Shapley value $s_i$ for a negative sample $d_i^-$ is defined as:*

$$s_i = H \sum_{\mathcal{Q} \subseteq \mathcal{D}^- - \{d_i^-\}} \frac{M(\mathcal{D}^+ \cup \mathcal{Q} \cup \{d_i^-\}, F) - M(\mathcal{D}^+ \cup \mathcal{Q}, F)}{\binom{N^- - 1}{|\mathcal{Q}|}} \tag{1}$$

*where $H$ is a constant, and the summation is taken over all subsets of negative samples, except $d_i^-$.*

Equation 1 can be re-expressed in the following form:

$$s_i = E_{\pi \sim \Pi}[M(\mathcal{D}^+ \cup A_\pi^{d_i^-} \cup \{d_i^-\}, F) - M(\mathcal{D}^+ \cup A_\pi^{d_i^-}, F)] \tag{2}$$

where $\Pi$ represents a uniform distribution of all the permutations among $\mathcal{D}^-$, and $A_\pi^{d_i^-}$ denotes all the negative samples before $d_i^-$ in a permutation $\pi$. Given the exponential complexity of computing the data Shapley values for negative samples , we further adopt the Monte Carlo permutation sampling technique to approximate the computation of $s_i$ (Castro et al., 2009). By repeating this approximation over multiple Monte Carlo permutations, we efficiently derive the estimated data Shapley value $s_i$. After computing the data Shapley value of each negative sample, we proceed to define the Negative Sample Shapley Field.

**Definition 1** *(Negative Sample Shapley Field) We define the Negative Sample Shapley Field $\mathcal{S}$ as an inherently existing scalar field representing the distribution of data Shapley values across all negative samples in space. In this field, each point denotes a negative sample $d_i^-$ and is associated with its data Shapley value $s_i$. Therefore, $\mathcal{S}$ is a mathematical function that maps the input of each negative sample to its corresponding data Shapley value: $\mathbf{x}_i \mapsto s_i$.*

With this field $\mathcal{S}$ constructed, our goal of cohort discovery within negative samples can be reframed as the task of identifying "hot zones" - grouped regions within $\mathcal{S}$ exhibiting high data Shapley values.

## 3.2 MANIFOLD LEARNING WITH STRUCTURE PRESERVATION AND ISOTROPY CONSTRAINT

As in Figure 1(a) and (b), although we hope to detect a similarly clustered cohort in the Negative Sample Shapley Field in both scenarios, the anisotropic nature of the space, i.e., the non-uniform distribution of negative samples with similar data Shapley values, presents significant challenges. To mitigate these challenges, we propose to employ manifold learning (Bengio et al., 2013) to transform the original space $\mathcal{S}$ into a new geometric space $\mathcal{S}'$. As elaborated in Sec. 2, to avoid mis-discovery such as Figure 1(b), we should simultaneously preserve the underlying structural information in the data while imposing an isotropy constraint on the data Shapley values in $\mathcal{S}'$. The resulting $\mathcal{S}'$ will be more amenable to accurate identification of medically relevant cohorts.

We employ a stacked denoising autoencoder (SDAE) (Vincent et al., 2010) as the backbone model for manifold learning due to its capability of handling input data corruption. Further, we integrate the isotropy constraint while preserving the data structure information in $\mathbf{x}_i$. Consider an SDAE consisting of $K$ denoising autoencoders (DAEs). For the $k$-th DAE ($k \in \{0, \ldots, K-1\}$), the encoder takes $\mathbf{h}_i^{(k)}$ as input, where $\mathbf{h}_i^{(0)} = \mathbf{x}_i$ is the original input. We define $\tilde{\mathbf{h}}_i^{(k)}$ as the corrupted version of $\mathbf{h}_i^{(k)}$ with masking noise generated by a stochastic mapping, $\tilde{\mathbf{h}}_i^{(k)} \sim g_\mathcal{D}(\tilde{\mathbf{h}}_i^{(k)} | \mathbf{h}_i^{(k)})$, which randomly sets a fraction of the elements of $\mathbf{h}_i^{(k)}$ to 0. The encoder transforms the corrupted $\tilde{\mathbf{h}}_i^{(k)}$ into an abstract representation $\hat{\mathbf{h}}_i^{(k+1)}$, which is then used by the decoder to recover the uncorrupted $\mathbf{h}_i^{(k)}$. This process equips the DAE with the capability of extracting useful information for denoising, which is crucial for healthcare analytics, due to missing data and noise in real-world EMR data (Lasko et al., 2013). The model architecture is depicted in Figure 6 of Appendix D.1.

**Encoder of the $k$-th DAE.** The encoder of the $k$-th DAE transforms the corrupted representation using an affine transformation followed by a non-linear activation function:

$$\hat{\mathbf{h}}_i^{(k+1)} = f_\theta^{(k+1)}(\tilde{\mathbf{h}}_i^{(k)}) = \sigma(\mathbf{W}_\theta^{(k+1)} \tilde{\mathbf{h}}_i^{(k)} + \mathbf{b}_\theta^{(k+1)}) \tag{3}$$

where $f_\theta^{(k+1)}(\cdot)$ is the encoder with $\mathbf{W}_\theta^{(k+1)}$ and $\mathbf{b}_\theta^{(k+1)}$ as the weight matrix and bias vector, respectively. The rectified linear unit (ReLU) activation function $\sigma(\cdot)$ is used for non-linearity.

**Decoder of the $k$-th DAE.** The derived abstract representation $\hat{\mathbf{h}}_i^{(k+1)}$ is subsequently mapped back to the previous latent space in the decoder, with the aim of recovering the uncorrupted representation:

$$\mathbf{z}_i^{(k)} = f_\phi^{(k+1)}(\hat{\mathbf{h}}_i^{(k+1)}) = \sigma(\mathbf{W}_\phi^{(k+1)}\hat{\mathbf{h}}_i^{(k+1)} + \mathbf{b}_\phi^{(k+1)}) \tag{4}$$

where $f_\phi^{(k+1)}(\cdot)$ is the decoder of the $k$-th DAE, with $\mathbf{W}_\phi^{(k+1)}$, $\mathbf{b}_\phi^{(k+1)}$ and the ReLU activation.

**Structure Preservation.** To attain a stable and robust abstract representation that is resilient to data corruption, it is crucial to recover the uncorrupted representation as accurately as possible. To achieve this, we adopt a reconstruction loss that preserves the data structure information. For a given batch of negative samples $\mathcal{B}$, the reconstruction loss per sample within this batch is defined as:

$$\mathcal{L}_{rec}^{(k)} = \frac{1}{|\mathcal{B}|} \sum_{i \in \mathcal{B}} \|\mathbf{h}_i^{(k)} - \mathbf{z}_i^{(k)}\|^2 \tag{5}$$

**Isotropy Constraint.** In addition to the reconstruction loss, it is essential to enforce an isotropy constraint to ensure that data Shapley value changes are uniform across orientations. To achieve this, we introduce a penalty that accounts for the change in data Shapley values relative to the Euclidean distance between two samples:

$$\mathcal{L}_{iso}^{(k)} = \frac{1}{|\mathcal{B}|^2} \sum_{i,j \in \mathcal{B}} \left(\frac{s_j - s_i}{\mu_{ij}}\right)^2 \tag{6}$$

where $i, j$ are two samples with $s_i$, $s_j$ as their data Shapley values, $\mu_{ij}$ as the distance between $\hat{\mathbf{h}}_i^{(k+1)}$ and $\hat{\mathbf{h}}_j^{(k+1)}$. The overall loss is then a weighted sum of the reconstruction loss and the isotropy penalty, jointly integrating the structural information and the isotropy constraint:

$$\mathcal{L}^{(k)} = \omega_{rec}\mathcal{L}_{rec}^{(k)} + \omega_{iso}\mathcal{L}_{iso}^{(k)} \tag{7}$$

The weights $\omega_{rec}$ and $\omega_{iso}$ are introduced to address the issue of the two loss terms being on different scales, which ensures that both losses are decreased at similar rates, leading to a better balance between the optimization objectives (Groenendijk et al., 2021; Liu et al., 2019). Specifically, the weights are set to the loss ratio between the current iteration ($t$) and the previous iteration ($t - 1$):

$$\omega_{rec} = \mathcal{L}_{rec}^{(k)}(t)/\mathcal{L}_{rec}^{(k)}(t-1), \quad \omega_{iso} = \mathcal{L}_{iso}^{(k)}(t)/\mathcal{L}_{iso}^{(k)}(t-1) \tag{8}$$

We have introduced how to learn the $k$-th DAE using the loss function in Equation 7. The corrupted input is only used during the initial training to learn robust feature extractors. After the encoder $f_\theta^{(k+1)}(\cdot)$ is trained, it will be applied to the clean input:

$$\mathbf{h}_i^{(k+1)} = f_\theta^{(k+1)}(\mathbf{h}_i^{(k)}) = \sigma(\mathbf{W}_\theta^{(k+1)}\mathbf{h}_i^{(k)} + \mathbf{b}_\theta^{(k+1)}) \tag{9}$$

$\mathbf{h}_i^{(k+1)}$ is used as input for the $(k + 1)$-th DAE to continue the repeated training process. When the last DAE is trained, we obtain the encoded representation $\mathbf{h}_i^{(K)}$ in the manifold space $\mathcal{S}'$, which preserves the data structure information in $\mathbf{x}_i$ and integrates the desired isotropy constraint.

### 3.3 COHORT DISCOVERY AMONG HIGH DATA SHAPLEY VALUE NEGATIVE SAMPLES

We proceed to perform cohort discovery in the encoded manifold space $\mathcal{S}'$, where each negative sample's input $\mathbf{x}_i$ is transformed into $\mathbf{h}_i^{(K)}$. We begin by setting a threshold value $\tau$ to filter out negative samples with data Shapley values below $\tau$, which focuses our analysis on negative samples with high data Shapley values, i.e., high contributions to the prediction task. Among the remaining negative samples with high data Shapley values, we target to detect the hot zones in $\mathcal{S}'$, which may represent medically meaningful cohorts of arbitrary shape.

To achieve this, we employ DBSCAN, short for density-based spatial clustering of applications with noise (Ester et al., 1996; Gan & Tao, 2015; Schubert et al., 2017) on such samples. The core idea of DBSCAN is to group samples that are close to each other in the manifold space $\mathcal{S}'$ into clusters, which could locate potential cohorts, whereas treating the remaining samples as noise or outliers. DBSCAN has three main steps: (i) identify the points within each point's $\varepsilon$- neighborhood and determine the "core points" with over $P_{min}$ neighbors; (ii) detect the connected components of the core points in the neighbor graph, disregarding any non-core points; (iii) assign each non-core

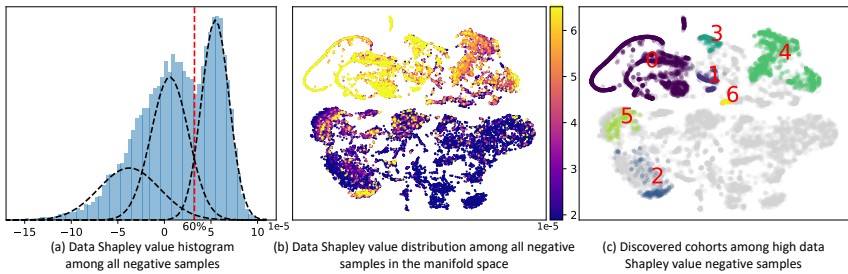

(a) Data Shapley value histogram among all negative samples

(b) Data Shapley value distribution among all negative samples in the manifold space

(c) Discovered cohorts among high data Shapley value negative samples

Figure 2: Cohort discovery of our proposal for AKI prediction.

point to the clusters which are the $\varepsilon$-neighborhood of the point; otherwise, label the point as noise. This process results in a set of clusters $\{C_1, C_2, \ldots, C_R\}$ and a set of noisy samples $\Psi$. Given the clusters, we define cohorts as follows.

**Definition 2** *(Cohorts) For a dense cluster $C_r$ identified by DBSCAN, we consider each of its core points and define a spherical space with the core point as its center and $\varepsilon$ as its radius. The joint space of all such spherical spaces is the cohort we aim to discover from this cluster.*

These discovered cohorts provide a promising avenue for further exploration of medically meaningful patterns in EMR data analytics, potentially revealing important insights.

## 4 EXPERIMENTAL EVALUATION

### 4.1 COHORT DISCOVERY IN CLINICAL VALIDATION

We evaluate our proposal using our hospital's EMR data to predict whether a patient will develop AKI in two days (as defined in Sec. 2). We employ the logistic regression (LR) model to compute the data Shapley value for each negative sample, using the area under the ROC curve (AUC) as the evaluation metric. More implementation details are elaborated in Appendices C.3, D.2 and E.3.

We present the cohort discovery results in Figure 2, where we first display the data Shapley value histogram among all the negative samples in Figure 2(a). It is noteworthy that this histogram can be well fitted by a Gaussian mixture model, consisting of three distinct and interesting components. The first component on the left represents the negative samples with negative data Shapley values. These samples have a negative impact on the prediction task, meaning that they are detrimental to predicting the AKI occurrence. In prior studies, one generally plausible explanation for the presence of such samples is the existence of mislabeled data (Ghorbani & Zou, 2019). However, for a representative acute disease like AKI, these negative samples are highly likely to be positive samples in the future but have not yet exhibited symptoms of AKI within the monitored time duration. Moving on to the second component in the middle, we observe that its data Shapley values are centered around a mean value close to zero. This implies that these negative samples are generally healthy without any apparent AKI-related symptoms. Notably, these healthy samples constitute a relatively significant portion of the data, which is commonly observed in clinical practice and aligns with our initial expectations. The third component on the right represents negative samples that are particularly valuable for the prediction task and merit special attention in our study. To further investigate these samples, we introduce a separation line between the second and third components, i.e., a threshold $60\%$ to exclude the lower $60\%$ negative samples based on their data Shapley values while retaining the remaining $40\%$ for further analysis. Our focus is on these remaining $40\%$ samples for identifying the hot zones, as illustrated in Figure 1.

The distribution of all negative samples, in terms of their data Shapley values in the manifold space, is presented in Figure 2(b). Upon performing DBSCAN on the extracted $40\%$ samples with high data Shapley values (points brighter than dark blue), we identify seven distinct cohorts of interest that are visually displayed using t-SNE plots in Figure 2(c), where grey points are either with low data Shapley values or labeled as noise by DBSCAN. We observe that these discovered cohorts are distinguishable from one another, potentially corresponding to medically meaningful patterns.

### 4.2 IN-DEPTH ANALYSIS OF DISCOVERED COHORTS

**Cohort 2: inflammatory cohort.** Figure 3(a) indicates a pronounced neutrophil-to-lymphocyte ratio (NLR) (Zahorec et al., 2001) in this patient group, marked by an increase in neutrophils (FNM)

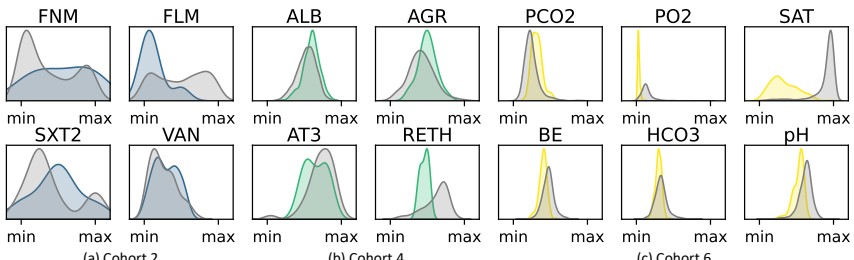

Figure 3: Lab test patterns of discovered Cohorts 2, 4, and 6. In each cohort, the colored region (blue, green, and yellow) represents the lab test value probability density of the samples in the cohort, while the grey region denotes that of all the other samples outside the cohort.

and a decrease in lymphocytes (FLM). This pattern, often tied to infectious, inflammatory, and stress conditions, suggests an overactive immune response leading to reduced lymphocyte counts (Nathan, 2006; Dhabhar, 2009). An elevated NLR, a reliable inflammatory marker, indicates a propensity for invasive infections (Huang et al., 2020b). Meanwhile, the levels of Cotrimoxazole (SXT2) and Vancomycin (VAN), both administered to treat infections including those associated with methicillin-resistant staphylococcus (Holmes & Howden, 2014), are found to be elevated in the bodies of these patients. The findings suggest that this patient cohort comprises individuals experiencing infections and acute inflammation, and receiving antibiotic treatment. Severe infections can cause systemic inflammatory response syndrome and kidney injury. Antibiotics like vancomycin can worsen kidney stress and have nephrotoxic properties (Wu & Huang, 2018), potentially leading to kidney dysfunction during treatment. However, modern medical practice can effectively manage these cases. Infections are promptly treated with broad-spectrum antibiotics and at appropriate doses within safety limits; hence, the patients do not develop significant AKI (Goldstein et al., 2016).

**Cohort 4: hepatic and hematological disorders cohort.** As delineated in Figure 2(c), Cohort 4 exhibits an augmented region and an increased quantity of sampling points, indicative of a more expansive patient population. A comprehensive analysis of the lab test indicator distribution for this cohort, portrayed in Figure 3(b), reveals differences in levels of serum proteins. Specifically, derangements in levels of albumin (ALB) and the albumin-globulin ratio (AGR) signify aberrant protein synthesis in patients. These may be associated with hepatic dysfunction or hematological diseases such as myeloma (Spinella et al., 2016; Laudin et al., 2020). Hepatic diseases can lead to impaired production of other proteins such as antithrombin III (AT3) (Knot et al., 1984); AT3 may also be lost excessively in nephrotic syndrome which is a kidney disorder (Kauffmann et al., 1978), or undergo accelerated consumption in disseminated intravascular coagulation (Mammen, 1998). Diminished reticulocyte hemoglobin (RETH) is associated with iron deficiency anemia (Auerbach et al., 2021), and could either be linked to hematological disorders or nutritional deficiency. In addition, imbalances in albumin and globulin may also be associated with dehydration. Therefore, our observation derived from Cohort 4 may support the pathophysiological relationship that exists between disorders of the hematological and hepatic systems, which increases the propensity for kidney disease. Clinicians should exercise vigilance in care when managing these cases.

**Cohort 6: respiratory failure and metabolic acidosis cohort.** Figure 3(c) reveals significant metabolic imbalances in patients, leading to an acid-base imbalance. Specifically, increased carbon dioxide pressure (PCO2), reduced oxygen pressure (PO2), and insufficient blood oxygen saturation (SAT) suggest respiratory failure (Breen, 2001). Concurrently, reduced base excess (BE), bicarbonate ion (HCO3) levels, and blood pH values hint at metabolic acidosis, indicating possible acute illnesses causing lactic or ketoacidosis (Kraut & Madias, 2010). These results suggest potential severe respiratory complications, such as advanced pneumonia, heart failure-induced pulmonary edema, or chronic obstructive pulmonary disease (COPD) (Kempker et al., 2020). Alternatively, acute conditions like hypoxia, shock, or severe infection could disrupt aerobic metabolism, leading to anaerobic glucose conversion to lactate, which accumulates in the bloodstream and causes acidosis. This puts significant strain on the kidneys, potentially resulting in renal disease symptoms (Kraut & Madias, 2014). This cohort of patients under examination does not advance to AKI, leading to the inference that renal dysfunction may not constitute an end-organ complication. Rather, this patient cohort appears to exhibit a heightened disposition to respiratory failure.

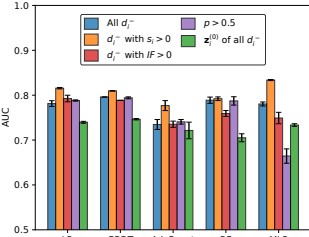 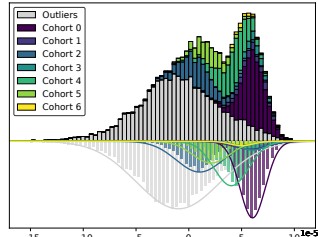

Figure 4: AKI prediction performance for our proposal against $IF$ and $p$ based data valuation.

Figure 5: Data Shapley value histogram of the samples within our discovered cohorts.

### 4.3 VALIDATION OF EFFECTIVENESS OF EACH COMPONENT

We validate the effectiveness of each component in our approach for AKI prediction. Specifically, we evaluate three settings of the negative sample usage in the training data (with positive samples the same): (i) all $d_i^-$: use all negative samples; (ii) $d_i^-$ with $s_i > 0$: only use the negative samples with positive data Shapley values; (iii) $\mathbf{z}_i^{(0)}$ of all $d_i^-$: use the decoded representations from the SDAE-based manifold learning. $\mathbf{z}_i^{(0)}$ is in the same dimension as the raw input but is in the decoding space after transformation by SDAE. We further compare with two data valuation baselines: (iv) $d_i^-$ with $IF > 0$: use the negative samples with positive $IF$ values, where $IF$ denotes influence functions measuring how the model changes when a single sample's weight is altered slightly (Weisberg & Cook, 1982); (v) $p > 0.5$: use the negative samples for which the prediction model is confident. We evaluate several widely adopted classifiers: LR, gradient-boosting decision tree (GBDT), adaptive boosting (AdaBoost), random forest (RF), and multilayer perceptron (MLP). The experimental results in AUC (mean ± std) from five repeats are illustrated in Figure 4.

**Effectiveness of Data Shapley Values for Negative Samples.** By comparing (i) and (ii), it is clear that after removing negative samples with data Shapley values smaller than $0$, all the classifiers exhibit an improvement in AUC. This substantiates the rationale behind our approach of associating samples of significant medical concern with their respective data Shapley values. Further, the efficacy of approximating data Shapley values through Monte Carlo permutation sampling is validated. In comparing data valuation methods, "$d_i^-$ with $IF > 0$" underperforms "$d_i^-$ with $s_i > 0$", as $IF$-based data valuation deletes informative negative samples, limiting its effectiveness in AKI prediction. Further, $IF$ fails to satisfy equitability conditions due to its inability to consider complex sample interactions, deviating from our focus on cohort discovery and posing robustness issues (Ghorbani et al., 2019). "$p > 0.5$" also performs less effectively than "$d_i^-$ with $s_i > 0$"as it excludes low-confidence samples, exacerbating overfitting. Hence, both $IF$ and $p$ based data valuations are unsuitable for AKI prediction. Detailed analyses of these comparisons are in Appendices G.4 and G.5.

**Effectiveness of Manifold Learning.** By changing the input data from the raw space to the decoder's output space after our proposed SDAE-based manifold learning (settings (i) vs. (iii)), we observe a moderate decrease in AUC, approximately $5\%$ in most classifiers. This decrease aligns with our expectations, as the transformation in SDAE introduces a certain level of information loss. However, the performance degradation remains within an acceptable range. These findings demonstrate that our proposed manifold learning manages to preserve the original data structure information and effectively model the original raw data space, despite a significant reduction in data dimension from 709 to 64. Thus, this corroborates our design rationale of employing SDAE for manifold learning with structure preservation and isotropy constraint. Further, we compare with contrastive principal component analysis (cPCA) (Abid et al., 2017; 2018) and our proposal without isotropy constraint, with results detailed in Appendices G.2 and G.1. Both methods primarily identify one large cohort and a few smaller cohorts, failing to identify medically meaningful cohorts. This limitation is also observed in deep clustering methods like deep clustering network (DCN) and deep embedded K-means clustering (DEKM), as discussed in Appendix G.6. Common to these methods is a process entailing dimensionality reduction followed by clustering on the embedded representations. However, their lack of our proposed isotropy constraint - crucial for uniform changes in data Shapley values across orientations - limits their efficacy. This is because they overlook the key insight that negative samples indicate different types with distinct symptoms, implying diverse cohorts with varied data Shapley value distributions. Consequently, they cannot effectively discover medically relevant cohorts.

**Effectiveness of Cohort Discovery.** We further validate the ability of our approach to decompose high data Shapley value samples into distinct, medically relevant cohorts. Figure 5 presents the data Shapley value histogram of our identified cohorts, with the upper part aligned with Figure 2(a) but color-coded by cohort proportion. The lower part shows each cohort's data Shapley value distribution. We note seven cohorts effectively partition Figure 2(a)'s third component into Gaussian distributions, implying consistent data Shapley values within each cohort. Cohort 2, identified as the inflammatory group, exhibits relatively lower data Shapley values, as immune abnormalities cannot serve as specific features for kidney injury. Conversely, Cohorts 4 and 6, involving critical metabolic systems, display higher data Shapley values, which indicates their significant medical relevance to AKI prediction. These observations confirm the homogeneity in each cohort due to DBSCAN's detection capability and similarity in data Shapley values, further substantiating our proposed isotropy constraint in manifold learning. Clinically validated by medical professionals, our derived cohort discovery results validate the correctness of the outcomes and the medical utility of our approach.

## 5 RELATED WORK

The Shapley value, originally introduced in cooperative game theory (Shapley et al., 1953), offers a solution for the equitable distribution of a team's collective value among its individual members (Chalkiadakis et al., 2011). Notable applications of the Shapley value in machine learning encompass data valuation, feature selection, explainable machine learning, etc (Rozemberczki et al., 2022; Ghorbani & Zou, 2019; Williamson & Feng, 2020; Lundberg & Lee, 2017; Liu et al., 2022). Among these, data valuation holds particular significance in quantifying the contributions of individual data samples toward predictive models. In this research line, the data Shapley value (Ghorbani & Zou, 2019) presents an equitable valuation framework for data value quantification with subsequent research focusing on enhancing computational efficiency (Jia et al., 2019; Ghorbani et al., 2020).

Representation learning is a crucial research area contributing to the success of many machine learning algorithms (Bengio et al., 2013). Among the representation learning methods, manifold learning stands out due to its capability of reducing the dimensionality and visualizing the underlying structure of the data. Traditional manifold learning methods include Isomap (Tenenbaum et al., 2000), locally linear embedding (Roweis & Saul, 2000), and multi-dimensional scaling (Borg & Groenen, 2005). In recent years, autoencoders (AEs) have gained significant attention in representation learning, offering efficient and effective representations of unlabeled data. Researchers develop various AE variants for specific application scenarios, among which DAEs and their advanced stacked variant SDAEs (Vincent et al., 2010) are highly suitable to tackle EMR data, in which missing and noisy data remains a notorious issue (Lasko et al., 2013).

DBSCAN, short for density-based spatial clustering of applications with noise, is introduced to alleviate the burden of parameter selection for users, facilitate the discovery of arbitrarily-shaped clusters, and demonstrate satisfactory efficiency when dealing with large datasets (Ester et al., 1996; Gan & Tao, 2015; Schubert et al., 2017).

## 6 CONCLUSION

This paper proposes to examine negative samples for cohort discovery in healthcare analytics, which has not been explored in prior research. In pursuit of this goal, we delve into an innovative, Shapley-based approach to uncover interrelationships among these samples, positing that cohorts of medical significance should manifest similar distributions with high data Shapley values. In particular, we propose to measure each negative sample's contribution to the prediction task via its data Shapley value and construct the Negative Sample Shapley Field to model the distribution of all negative samples. To enhance the cohort discovery quality, we transform this original field into an embedded space using manifold learning, incorporating the original data structure information and isotropy constraint. In the transformed space, we manage to identify medically meaningful cohorts within negative samples by DBSCAN. The experiments on our hospital's EMR data empirically demonstrate the effectiveness of our proposal. Further, the medical insights derived from our discovered cohorts are validated by clinicians, underscoring the substantial medical value of our approach.

## 7 REPRODUCIBILITY STATEMENT

In this paper, we strive to enhance the reproducibility of our proposed approach. We have discussed our approach comprehensively in Sec. 3, with a step-wise presentation. For the central component of our proposal, i.e., Negative Sample Shapley Field Construction, we have elucidated the necessary proofs of data Shapley values for negative samples in Appendix C.1 and introduced how we calculate data Shapley values for negative samples using Monte Carlo Permutation Sampling in Appendix C.2. As for our empirical results, we have listed the complete implementation details of each component within our proposal, namely Appendices C.3, D.2 and E.3 for Negative Sample Shapley Field Construction, Manifold Learning with Structure Preservation and Isotropy Constraint and Cohort Discovery Among High Data Shapley Value Negative Samples, respectively. Finally, we have included our source codes in the supplementary materials (i.e., the zip file).

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

# A  NOTATION TABLE

In this paper, scalars are denoted by symbols such as $x$, vectors are represented by boldface symbols such as $\mathbf{x}$, and matrices are described by uppercase boldface symbols such as $\mathbf{X}$. To provide a comprehensive overview of the notations used throughout the paper, we present a summary of notations in Table 1.

Table 1: Notations

| Notation | Description |
|----------|-------------|
| $\mathcal{D}, d_i$ | EMR data, each sample in EMR data |
| $\mathcal{D}^+, \mathcal{D}^-$ | Positive samples, negative samples |
| $d_i^-$ | Each negative sample |
| $\mathbf{x}_i, y_i$ | Input features of $d_i^-$ , binary label of $d_i^-$ |
| $F$ | Prediction model |
| $\mathcal{Q}$ | A subset of negative samples |
| $M$ | Performance metric function |
| $s_i$ | Data Shapley value for $d_i^-$ |
| $\pi$ | A Monte Carlo permutation |
| $A_\pi^{d_i^-}$ | All the negative samples before $d_i^-$ in $\pi$ |
| $\mathcal{S}$ | The Negative Sample Shapley Field |
| $\mathcal{S}'$ | Transformed space after SDAE-based manifold learning |
| $K$ | Number of DAEs in SDAE |
| $k$ | Each DAE in SDAE, $k \in \{0, \dots, K-1\}$ |
| $\mathbf{h}_i^{(k)}$ | Input to the encoder of the $k$-th DAE |
| $\tilde{\mathbf{h}}_i^{(k)}$ | Corrupted version of $\mathbf{h}_i^{(k)}$ with masking noise |
| $f_\theta^{(k+1)}(\cdot)$ | Encoder of the $k$-th DAE |
| $\hat{\mathbf{h}}_i^{(k+1)}$ | Output from the encoder of the $k$-th DAE |
| $f_\phi^{(k+1)}(\cdot)$ | Decoder of the $k$-th DAE |
| $\mathbf{z}_i^{(k)}$ | Output from the decoder of the $k$-th DAE |
| $\mathcal{L}_{rec}^{(k)}$ | Reconstruction loss in the $k$-th DAE |
| $\mathcal{L}_{iso}^{(k)}$ | Isotropy constraint in the $k$-th DAE |
| $\mathcal{L}^{(k)}$ | Overall loss in the $k$-th DAE |
| $\mathbf{h}_i^{(k+1)}$ | Input to the encoder of the $(k+1)$-th DAE |
| $\mathbf{h}_i^{(K)}$ | Input for medical cohort discovery |

## B    EXTENDED RELATED WORK

### B.1    COHORT STUDIES

Cohort studies, a specific subtype of longitudinal studies, focus on selecting a group of patients who share a common defining characteristic in order to investigate a particular outcome of interest. They represent a compelling research avenue within the realm of healthcare analytics (Wang & Kattan, 2020; Grimes & Schulz, 2002). Cohort studies are well-suited for identifying potential risk factors and causes, and monitoring the progression of diseases in patients' health conditions. For instance, in Mahmood et al. (2014), significant medical insights into the epidemiology of cardiovascular disease and its associated risk factors are provided through a cohort study. Another notable example pertains to the coronavirus disease 2019 (COVID-19), where multiple cohort studies have consistently demonstrated the critical role of D-dimer as a risk factor contributing to the mortality of COVID-19 patients (Wang et al., 2020; Zhou et al., 2020; Huang et al., 2020a). Leveraging cohort studies, researchers acquire the capacity to meticulously scrutinize various medical conditions, yielding invaluable medical insights. This, in turn, has the potential to drive substantial advancements in patient management and healthcare delivery (Wu et al., 2020; Szklo, 1998).

### B.2    RELATED WORK ON EACH COMPONENT OF OUR APPROACH

The Shapley value, originally introduced in cooperative game theory (Shapley et al., 1953), offers a solution for the equitable distribution of a team's collective value among its individual members (Chalkiadakis et al., 2011). This value allocation mechanism embodies key principles such as fairness, symmetry, and efficiency (Chalkiadakis et al., 2011), rendering it widely applicable across various machine learning applications (Rozemberczki et al., 2022). Notable applications of the Shapley value in machine learning encompass data valuation, feature selection, explainable machine learning, etc (Rozemberczki et al., 2022; Ghorbani & Zou, 2019; Williamson & Feng, 2020; Lundberg & Lee, 2017; Liu et al., 2022). Among these applications, data valuation holds particular significance in quantifying the contributions of individual data samples toward predictive models. In this research line, the data Shapley value (Ghorbani & Zou, 2019) presents an equitable valuation framework for data value quantification. Subsequent research efforts primarily focus on enhancing the computational efficiency of the data Shapley value through the application of specific techniques (Jia et al., 2019; Ghorbani et al., 2020).

Representation learning, i.e., learning data representations that benefit downstream tasks, is a crucial research area contributing to the success of many machine learning algorithms (Bengio et al., 2013). Among the representation learning methods, manifold learning, which operates under the assumption that the probability mass of the original data tends to concentrate in lower-dimensional regions compared to the original space, stands out due to its capability of reducing the dimensionality and visualizing the underlying structure of the data. Traditional manifold learning methods include Isomap (Tenenbaum et al., 2000), locally linear embedding (Roweis & Saul, 2000), and multidimensional scaling (Borg & Groenen, 2005). In recent years, AEs have garnered substantial interest in representation learning. AEs excel in capturing underlying data structures by reconstructing input data, thereby providing efficient and effective representations of unlabeled data. Researchers develop various AE variants for specific application scenarios, e.g., regularized AEs (Alain & Bengio, 2014), sparse AEs (Makhzani & Frey, 2014), DAEs (Vincent et al., 2008). For example, regularized AEs (Alain & Bengio, 2014) are proposed to prevent AEs from learning trivial identity mappings and to enhance their ability to capture comprehensive information from data. More specifically, sparse AEs (Makhzani & Frey, 2014), inspired by the sparse coding hypothesis in neuroscience, aim to learn sparse representations, and DAEs (Vincent et al., 2008) are introduced to learn representations robust to noise and outliers, hence effectively handling input data corruption. Specifically, DAEs and their advanced stacked variant SDAEs (Vincent et al., 2010) are highly suitable to tackle EMR data, in which missing and noisy data remains a notorious issue (Lasko et al., 2013). These models could effectively address the complexities associated with EMR data and contribute to improved representation learning.

DBSCAN, short for density-based spatial clustering of applications with noise, is introduced to alleviate the burden of parameter selection for users, facilitate the discovery of arbitrarily-shaped clusters, and demonstrate satisfactory efficiency when dealing with large datasets (Ester et al., 1996; Gan & Tao, 2015; Schubert et al., 2017). A subsequent study, $\rho$-approximate DBSCAN further

advances the quality of cluster approximation and computational efficiency (Gan & Tao, 2015). Then in Schubert et al. (2017), it is shown that the original DBSCAN algorithm, when equipped with appropriate indexes and parameters, can achieve performance comparable to that of the $\rho$-approximate DBSCAN algorithm. Up till now, DBSCAN remains one of the most widely adopted clustering algorithms.

### B.3  RELATED WORK ON BASELINE METHODS

Contrastive principal component analysis (cPCA) is a generalized variant of the standard PCA. Its primary purpose is to visualize and investigate patterns specific to a target dataset in contrast to an existing background dataset. In this manner, cPCA excels at identifying crucial dataset-specific patterns that might be missed by PCA (Abid et al., 2017; 2018).

In the setting of learning from positive and unlabeled data, generally referred to as PU learning, we only have access to positive examples and unlabeled data for analytics. PU learning has garnered increasing interest within the machine learning community, and among the notable research endeavors, three influential PU learning methods have emerged: Classic Elkanoto (Elkan & Noto, 2008), Weighted Elkanoto (Elkan & Noto, 2008), and Bagging-based PU-learning (Mordelet & Vert, 2014). The first two methods are founded on the assumption of samples being "selected completely at random," while the latter, Bagging-based PU-learning, leverages bootstrap aggregating (bagging) techniques to achieve improved performance.

## C    Negative Sample Shapley Field Construction

### C.1    Proof of Data Shapley Values for Negative Samples

We establish the proof of data Shapley values for negative samples by relating our problem to the original context of the Shapley value in game theory (Shapley et al., 1953), i.e., reducing it to a cooperative game (Rozemberczki et al., 2022; Ghorbani & Zou, 2019; Chalkiadakis et al., 2011).

Specifically, our problem is framed as a negative sample valuation game for a fair distribution of the collective performance achieved by the prediction model to each participating negative sample in the training data (with positive samples the same), while maintaining consistency with the three fundamental properties of an equitable data valuation: (i) null player, (ii) symmetry, and (iii) linearity.

**Null player.** We define a negative sample $d_i^-$ as a "null player" and set its data Shapley value to zero if its inclusion in any subsets of the negative sample set in training data does not influence the performance of the prediction model. Formally, for a negative sample $d_i^- = (\mathbf{x}_i, y_i)$ and $\forall \mathcal{R} \subseteq \mathcal{D}^- \setminus d_i^-$, if the performance remains unchanged by adding $d_i^-$, i.e., $M(\mathcal{D}^+ \cup \mathcal{R}, F) = M(\mathcal{D}^+ \cup \mathcal{R} \cup \{d_i^-\}, F)$, then $s_i = 0$. In this negative sample valuation game, the null player property ensures that the negative samples with no impact on the prediction performance are assigned zero values for their data Shapley values.

**Symmetry.** Two negative samples, $d_i^-$, and $d_j^-$, are assigned the same value if they consistently influence the performance of the prediction model when added to any subsets of the negative sample set in training data. This property arises from the concept of symmetry. Formally, for two negative samples $d_i^- = (\mathbf{x}_i, y_i)$ and $d_j^- = (\mathbf{x}_j, y_j)$, and $\forall \mathcal{R} \subseteq \mathcal{D}^- \setminus \{d_i^-, d_j^-\}$, if the prediction performance remains the same after adding $d_i^-$ or $d_j^-$, i.e., $M(\mathcal{D}^+ \cup \mathcal{R} \cup \{d_i^-\}, F) = M(\mathcal{D}^+ \cup \mathcal{R} \cup \{d_j^-\}, F)$, then $s_i = s_j$. This property ensures that the negative samples with equivalent marginal contributions are assigned the same data Shapley values.

**Linearity.** The influence of a negative sample $d_i^-$ on the overall pooled data is equivalent to its influence on constituent sub-datasets. We could denote $s_i$ as $s_i(d_i^-, \mathcal{D}_{test})$, representing the data Shapley value of the negative sample $d_i^-$ evaluated on all test data $\mathcal{D}_{test}$. The linearity property states that for two sets of test data, $\mathcal{D}_{test}^1$ and $\mathcal{D}_{test}^2$, the following holds:

$$s_i(d_i^-, \mathcal{D}_{test}^1 \cup \mathcal{D}_{test}^2) = s_i(d_i^-, \mathcal{D}_{test}^1) + s_i(d_i^-, \mathcal{D}_{test}^2) \tag{10}$$

This linearity property ensures that the data Shapley value of a negative sample on the pooled test dataset is equal to the sum of its data Shapley values on the two individual test datasets, in this negative sample valuation game.

**Proof 1** *We prove Proposition 1 by establishing the connection between our negative sample valuation game and the cooperative game theory context (Chalkiadakis et al., 2011). In a cooperative game, there exists a set of $n$ players and a characteristic function $m : 2^{[n]} \mapsto \mathbb{R}$ that assigns a payment value to each selected player (Rozemberczki et al., 2022). In our case, the players correspond to individual negative samples, and the characteristic function $m(\mathcal{Q})$ represents the performance obtained when the subset of negative samples $\mathcal{Q}$ ($\mathcal{Q} \subseteq \mathcal{D}^-$) is included in the prediction model. By leveraging the three properties discussed above, our negative sample valuation game ensures the fair distribution of collective performance to the participating negative samples. Therefore, each negative sample acts as a player, and the prediction model $F$ incorporates all the participating negative samples $\mathcal{Q}$ (along with the positive samples $\mathcal{D}^+$) to achieve the overall performance $m = M(\mathcal{D}^+ \cup \mathcal{Q}, F)$. Consequently, the data Shapley value of each negative sample corresponds to the payment received by each player in this cooperative game analogy.* □

### C.2    Monte Carlo Permutation Sampling

We adopt Monte Carlo permutation sampling to approximate the data Shapley values for negative samples. The detailed procedure of each Monte Carlo iteration is presented in Algorithm 1. The algorithm begins by initializing the necessary variables for computation in lines 1-8. Subsequently, for a given permutation, we calculate the marginal contribution of each negative sample in the current Monte Carlo iteration towards its overall data Shapley value, as described in lines 9-25.

In particular, for each indexed negative sample, we include it in the training data and retrain the classifier. Then, we measure its marginal contribution by calculating the difference in the AUC metric (lines 10-14). Additionally, in line 15, we compute the absolute difference between the full AUC (which uses all the training data and evaluates the trained model on the test data) and the new AUC (which includes the current negative sample). If this difference falls below a predefined threshold, specifically "$truncation\_tolerance$" times the full AUC, for more than five consecutive negative samples, we terminate the current Monte Carlo iteration by early stopping (lines 16-21). This early stopping criterion is based on the observation that further inclusion of negative samples is unlikely to yield a significant improvement in AUC.

After calculating the marginal contribution of each negative sample in each Monte Carlo iteration, the overall data Shapley value of a particular negative sample is derived by taking the mean of its marginal contributions across different iterations.

---

**Algorithm 1:** Data Shapley Value Computation for Negative Samples by Monte Carlo Sampling

---

**Input** : Negative training data $(\mathbf{X}_{train}^-, \mathbf{y}_{train}^-)$, Positive training data $(\mathbf{X}_{train}^+, \mathbf{y}_{train}^+)$, Test data $(\mathbf{X}_{test}, \mathbf{y}_{test})$.
**Output:** The marginal contribution of each negative sample in the current Monte Carlo iteration to its overall data Shapley value.

1  Initialize permutation of indices of $\mathbf{X}_{train}^-$: $perm \leftarrow$ random permutation
2  Initialize marginal contributions of $\mathbf{X}_{train}^-$ with zeros: $marginal\_contribs \leftarrow$ zeros
3  Initialize truncation counter: $truncation\_counter \leftarrow 0$
4  Initialize new score with a random score: $new\_score \leftarrow$ random_score `// 0.5 for AUC`
5  Initialize a classifier: $clf \leftarrow$ create a new classifier
6  Fit the classifier with all training data: $clf$.fit($\mathbf{X}_{train}^- \cup \mathbf{X}_{train}^+, \mathbf{y}_{train}^- \cup \mathbf{y}_{train}^+$)
7  Evaluate the classifier on test data: $full\_score \leftarrow$ AUC($clf, \mathbf{X}_{test}, \mathbf{y}_{test}$)
8  Initialize training data: $(\mathbf{X}', \mathbf{y}') \leftarrow (\mathbf{X}_{train}^+, \mathbf{y}_{train}^+)$

9  **for** $idx$ **in** $perm$ **do**
10      Set old score to the current new score: $old\_score \leftarrow new\_score$
11      Update training data with current negative sample:
     $(\mathbf{X}', \mathbf{y}') \leftarrow (\mathbf{X}' \cup \mathbf{X}_{train}^-[idx], \mathbf{y}' \cup \mathbf{y}_{train}^-[idx])$
12      Create a new classifier and train it: $clf \leftarrow$ new classifier, $clf$.fit($\mathbf{X}', \mathbf{y}'$)
13      Update new score: $new\_score \leftarrow$ AUC($clf, \mathbf{X}_{test}, \mathbf{y}_{test}$)
14      Calculate the marginal contribution of the current negative sample:
     $marginal\_contribs[idx] \leftarrow new\_score - old\_score$
15      Calculate the distance to the full score: $distance\_to\_full\_score \leftarrow |$full_score - new_score$|$
16      **if** $distance\_to\_full\_score \leq truncation\_tolerance \times full\_score$ **then**
17         Increment truncation counter: $truncation\_counter \leftarrow truncation\_counter + 1$
18         **if** $truncation\_counter > 5$ **then**
19            **break**
20         **end**
21      **end**
22      **else**
23         Reset truncation counter: $truncation\_counter \leftarrow 0$
24      **end**
25  **end**

26  **return** $marginal\_contribs$

---

### C.3 IMPLEMENTATION DETAILS

In our experiments, we employ the LR model to approximate the data Shapley values for negative samples. We use AUC as the evaluation metric with the early stopping criterion as previously described. Specifically, we set the threshold $truncation\_tolerance$ to 0.025. This means that if the absolute difference between the full AUC and the new AUC remains within 0.025 times the full AUC for more than five consecutive negative samples, the current Monte Carlo iteration will be terminated.

# D  MANIFOLD LEARNING WITH STRUCTURE PRESERVATION AND ISOTROPY CONSTRAINT

## D.1  MODEL ARCHITECTURE OF SDAE-BASED MANIFOLD LEARNING

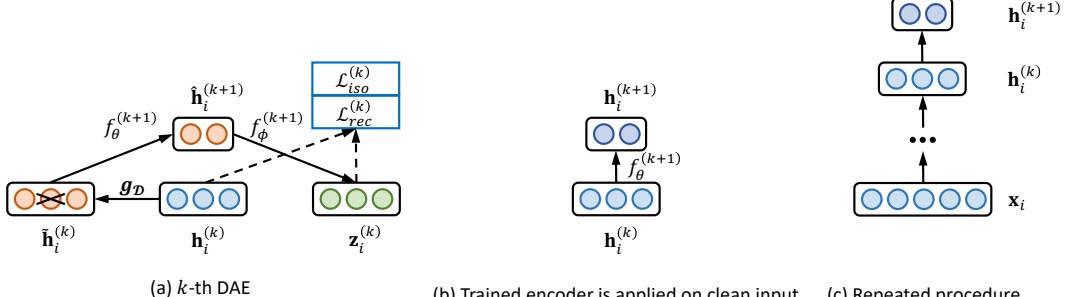

Figure 6: Model architecture of SDAE-based manifold learning.

The architecture of our SDAE-based manifold learning model is presented in Figure 6. Specifically, Figure 6 (a) illustrates the computation details of the $k$-th DAE (denoising autoencoder). Both the reconstruction loss and our proposed isotropy constraint are employed in this process, and hence contribute to the training of robust feature extractors, utilizing clean and corrupted inputs simultaneously. Subsequently, after training the encoder of the $k$-th DAE, it is applied to clean inputs for transformation, as depicted in Figure 6 (b). This iterative training process continues, as shown in Figure 6 (c), until the last DAE generates the final encoded representation within the manifold space. This encoded representation serves as the foundation for subsequent cohort discovery.

## D.2  IMPLEMENTATION DETAILS

We utilize an SDAE comprising 3 DAEs. These DAEs serve to transform the 709-dimension input data, which corresponds to 709 lab tests (refer to Appendix F.1), using encoders with dimensions of 256, 128, and 64, respectively. We utilize the Adam optimizer to train the SDAE in an unsupervised manner, using the loss function described in Equation 7. Our objective is to obtain an optimal manifold space to support subsequent density-based clustering for automatic cohort identification. To determine the optimal learning rate for training, we perform a grid search over a range of values, i.e., $[0.1, 0.05, 0.01, 0.005, 0.001, 0.0005]$, and run 10 repeats per learning rate. The model run with the lowest loss is selected as the optimal model for subsequent cohort discovery, which corresponds to a learning rate of $0.005$. Other parameters are held constant during the training process, including a batch size of $1024$, a mask probability of $0.2$ for the denoising process, and a total of 100 epochs. These parameters provide stability and ensure sufficient training iterations to learn meaningful representations in the SDAE.

## E    COHORT DISCOVERY AMONG HIGH DATA SHAPLEY VALUE NEGATIVE SAMPLES

### E.1    DETAILS FOR DBSCAN

The DBSCAN (density-based spatial clustering of applications with noise) algorithm follows a specific process to perform clustering. It requires two essential parameters: (i) $\varepsilon$, which defines the maximum distance between two samples for them to be considered neighbors, and (ii) $P_{min}$, which specifies the minimum number of samples required to form a dense region.

The detailed description of the DBSCAN algorithm is as follows. (i) Start by selecting an unvisited sample arbitrarily. (ii) Retrieve its $\varepsilon$-neighborhood, consisting of all samples within a distance of $\varepsilon$ from the selected sample. (iii) If the $\varepsilon$-neighborhood contains more than $P_{min}$ samples, initiate a new cluster and designate the selected sample as a "core point". The core point is a sample that has a sufficient number of neighbors within its $\varepsilon$-neighborhood to form a dense region. (iv) If the $\varepsilon$-neighborhood has fewer than $P_{min}$ samples, label the selected sample as noise. However, note that this sample may later fall within the $\varepsilon$-neighborhood of another sample, causing it to be assigned to a different cluster. (v) For each sample that is determined to belong to a dense region within a cluster, consider its $\varepsilon$-neighborhood as part of the same cluster. Add all the samples found within this neighborhood to the cluster and check if these samples' respective $\varepsilon$-neighborhoods are also dense (if so, add them to the cluster as well). This process continues recursively until the entire densely connected cluster is detected. (vi) Proceed to the next unvisited sample and repeat steps (ii) to (v) until all samples have been assigned to a cluster or labeled as noise. By following this process, DBSCAN identifies densely connected $\{C_1, C_2, \ldots, C_R\}$ and recognizes noisy samples $\Psi$.

### E.2    CLUSTERS VS. COHORTS

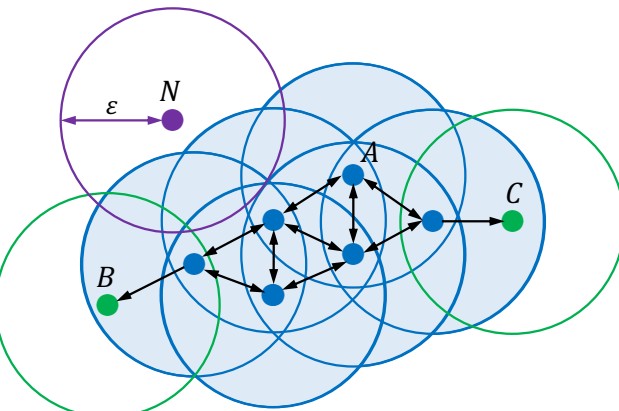

Figure 7: Relationship between clusters and cohorts in a DBSCAN example.

We illustrate the relationship between clusters and cohorts using an example within the DBSCAN algorithm, as depicted in Figure 7. In this example, we set $P_{min}$ to 4, and the value of $\varepsilon$ is indicated in the figure as the radius of the circles.

As shown in the figure, Point A and all the other blue points are core points because their $\varepsilon$-neighborhoods contain at least $P_{min}$ points. Therefore, they form a single cluster. Additionally, Point B and Point C are reachable from Point A via existing paths, making them belong to the same cluster as well. However, Point N is labeled as noise since it does not meet the criteria to be a core point and is not reachable from any core points.

According to Definition 2, for this identified cluster by the DBSCAN algorithm, we consider each core point (i.e., all the blue points) and define a spherical space with the core point as its center and $\varepsilon$ as its radius. The combined area covered by all such spherical spaces, depicted in blue, represents the cohort that we aim to discover from this cluster.

### E.3 IMPLEMENTATION DETAILS

The choice of parameters in the DBSCAN algorithm, specifically the search radius ($\varepsilon$) and the minimum number of points ($P_{min}$), has a significant impact on the quality of the clustering results. To determine the optimal parameter combination, we start by exploring various values for $P_{min}$.

Given a specific $P_{min}$ value, we calculate the 75th percentile of the distribution of ($P_{min}/2$)-nearest distances for the extracted 40% samples with high data Shapley values (as described in Sec. 4.1). We consider this calculated value as the appropriate $\varepsilon$ for the clustering process. The underlying rationale is that regions with local densities exceeding twice the upper bound of the global density represent distinct high-density areas.

By iterating over different values of $P_{min}$ and adjusting the corresponding $\varepsilon$ values, we assess the clustering quality achieved by each parameter combination using the Silhouette score, which measures the cohesion and separation of clusters to evaluate their quality. After a thorough evaluation, we determine that a value of $P_{min}$ equal to 100 yields the most suitable parameter choice for our DBSCAN clustering. This method ensures that the clustering process considers the distribution of distances within high-density areas and selects an appropriate value for $\varepsilon$, leading to improved clustering results based on the Silhouette score assessment.

# F  EXPERIMENTAL SET-UP

## F.1  HOSPITAL-ACQUIRED AKI PREDICTION AND DATA PROCESSING

Hospital-acquired AKI (short for acute kidney injury) is a disease we strive to handle in our medical practice as front-line clinicians and medical researchers. According to the KDIGO criteria (Kellum et al., 2012), the definition of AKI is based on the rise of sCr (i.e., serum creatinine), a lab test, beyond a threshold limit within a defined timeline. The definition includes two criteria: absolute AKI and relative AKI, as depicted in Figure 8. Absolute AKI is defined as an increase in sCr of more than 26.5 umol/L within the past two days. Relative AKI, on the other hand, is defined as a rise in sCr of 1.5 times or higher compared to the lowest sCr value within the last seven days.

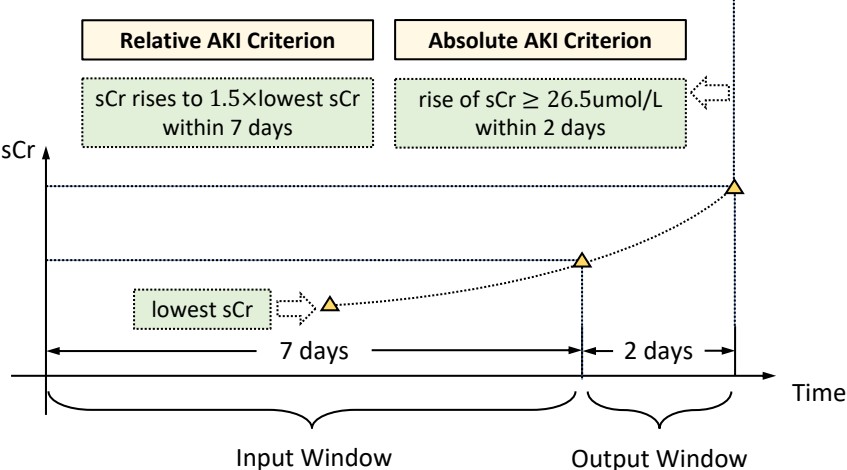

Figure 8: Definition of absolute AKI and relative AKI in hospital-acquired AKI prediction.

In hospital-acquired AKI prediction, our goal is to predict whether a patient will develop AKI within two days of hospital admission. We evaluate our approach on our hospital's EMR data, containing 709 lab tests as input features. Each hospitalized admission in the data is treated as a sample for analysis. In total, we receive 20,732 admissions, with 911 of them resulting in AKI development. We partition the dataset into 90% training data and 10% testing data.

For positive samples where AKI develops during admission, we record the time of AKI detection and define a two-day window, referred to as the "Output Window", that counts backward from the detection time. This window is not used as input but is crucial in medical practice as it provides a 48-hour lead time, enabling clinicians to take timely interventions following AKI prediction if necessary. The "Input Window", which serves as input for analysis, spans seven days prior to the Output Window. The relationship between the Input Window and the Output Window is depicted in Figure 8. For negative samples, the time of the last recorded lab test is used to determine both the Output Window and the Input Window, respectively.

In summary, our approach utilizes 709 lab tests within the Input Window to predict the likelihood of each sample (i.e., admission) developing AKI after the Output Window. We perform the min-max standardization on the lab test values and then calculate the average to derive input features. Table 2 presents key statistics of our dataset for hospital-acquired AKI prediction.

Table 2: Key statistics of our dataset for hospital-acquired AKI prediction

| Statistics | Our Dataset |
|---|---|
| # of admissions | 20732 |
| # of positive samples | 911 |
| # of negative samples | 19821 |
| # of lab tests | 709 |
| Input Window | 7 days |
| Output Window | 2 days |

## F.2 EXPERIMENTAL ENVIRONMENT

We conduct the experimental evaluation on a server with the specifications as follows. (i) CPU: Intel(R) Xeon(R) Gold 6248R × 2, with a clock speed of 3.0GHz and 24 cores per chip. (ii) Memory: the server is equipped with 768GB of memory. (iii) GPU: there are 8 NVIDIA V100 GPUs with NVLINK technology, providing a high-speed data transfer rate of 300GB/s. (iv) Software: The models are implemented using PyTorch version 1.12.1.

# G  SUPPLEMENTARY EXPERIMENTAL RESULTS

## G.1  EXPERIMENTAL RESULTS ON EFFECTIVENESS OF ISOTROPY CONSTRAINT

The comprehensive results of the ablation study, comparing the cohort discovery results of our proposal with and without the isotropy constraint, are presented in Figure 9, encompassing various $P_{min}$ settings. According to the comparison results, we observe that the results of our proposal (with the isotropy constraint) remain relatively stable across varied $P_{min}$ settings ($P_{min} = 100$ for Figure 2(c) as described in Appendix E.3). Notably, upon removing the isotropy constraint, effective cohort discovery through DBSCAN is impeded, irrespective of the chosen $P_{min}$ values.

This is because DBSCAN, as a spatial clustering method, necessitates appropriate handling of spatial information within the Negative Sample Shapley Field to achieve meaningful cohort discovery. Therefore, the introduced isotropy constraint plays a pivotal role by ensuring uniform changes in data Shapley values across orientations, thereby rendering our proposal more amenable to subsequent spatial clustering. Consequently, we mitigate the risk of mis-discovery, as exemplified in Figure 1(b), and ensure robust cohort discovery outcomes resilient to variations in spatial clustering algorithm parameters, ultimately contributing to unveiling medically meaningful cohorts.

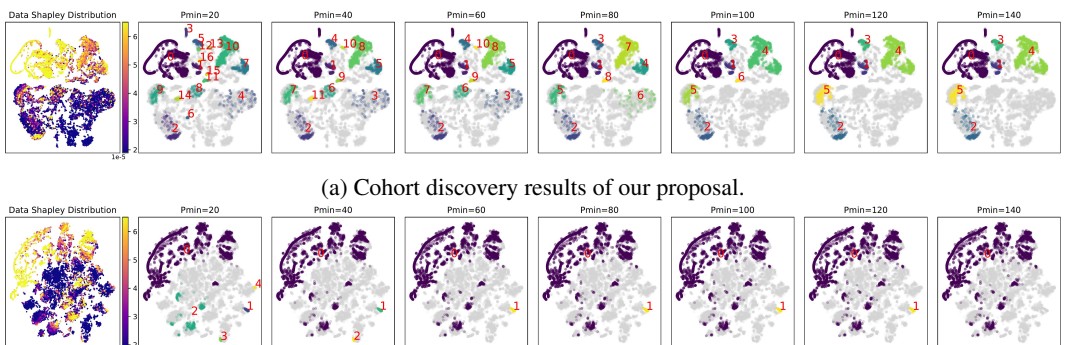

(a) Cohort discovery results of our proposal.

(b) Cohort discovery results of our proposal without isotropy constraint.

Figure 9: Comparison between our proposal with vs. without isotropy constraint in cohort discovery.

## G.2  COMPARISON WITH CPCA

cPCA, short for contrastive principal component analysis, represents a generalization of the standard PCA. By utilizing a background dataset to eliminate common patterns, cPCA's objective is to unveil the unique patterns within the target dataset relative to the background dataset (Abid et al., 2017; 2018). In this regard, cPCA shares a conceptual similarity with our proposal. Therefore, we proceed to conduct a comparative analysis between our approach and cPCA specifically for cohort discovery.

Considering our focus on identifying cohorts within negative samples, we construct the background dataset using positive samples, while the negative samples constitute the target dataset. Following the projection of the data through cPCA, we retain 64 contrastive principal components, a dimension that is consistent with the output of SDAE in our approach. Subsequently, we employ DBSCAN on the projected data resulting from cPCA to achieve cohort discovery. In essence, cPCA can be regarded as an embedding technique, serving as a counterpart to the combination of the first two components in our proposed approach: Negative Sample Shapley Field Construction and Manifold Learning with Structure Preservation and Isotropy Constraint.

Comprehensive experimental results of cPCA are depicted in Figure 10, presenting cohort discovery outcomes for four distinct $\alpha$ values: 0, 1.06, 5.54, and 74.44 (automatically determined by cPCA), across different $P_{min}$ settings. Comparing these cPCA results with our proposal's results illustrated in Figure 9(a), it is observed that our proposal demonstrates a clear superiority over cPCA in cohort discovery across different $\alpha$ values as well as varying $P_{min}$ settings.

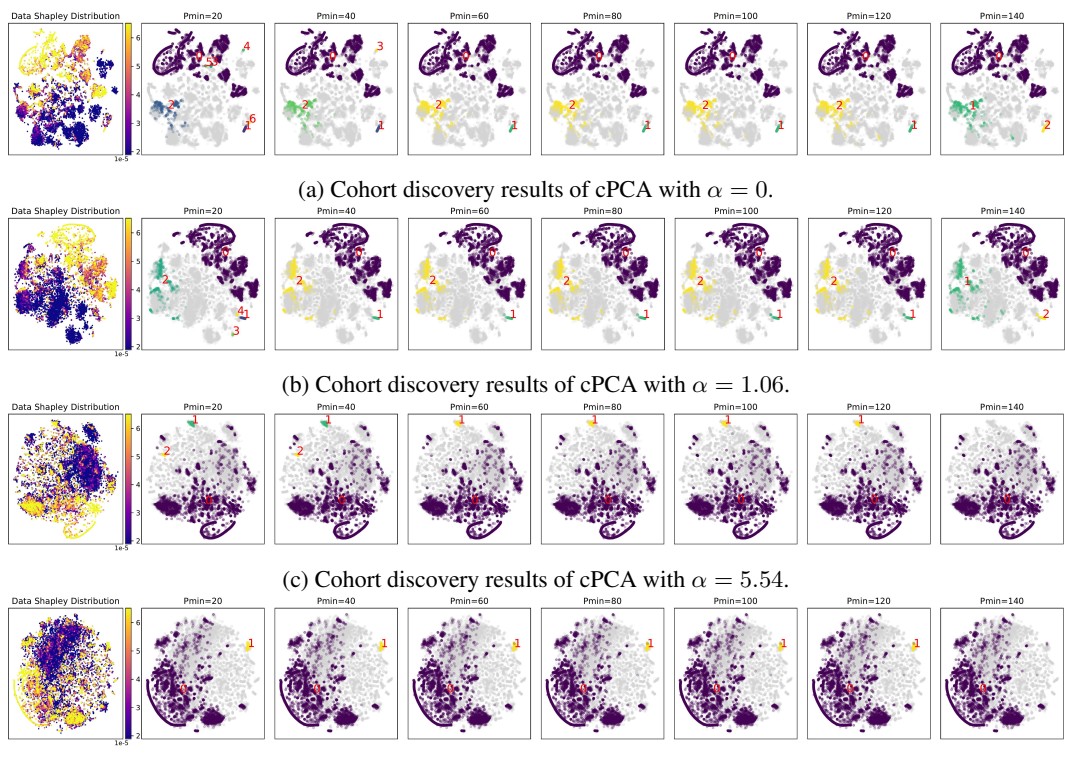

(a) Cohort discovery results of cPCA with $\alpha = 0$.

(b) Cohort discovery results of cPCA with $\alpha = 1.06$.

(c) Cohort discovery results of cPCA with $\alpha = 5.54$.

(d) Cohort discovery results of cPCA with $\alpha = 74.44$.

Figure 10: Cohort discovery results of cPCA with four different $\alpha$ settings.

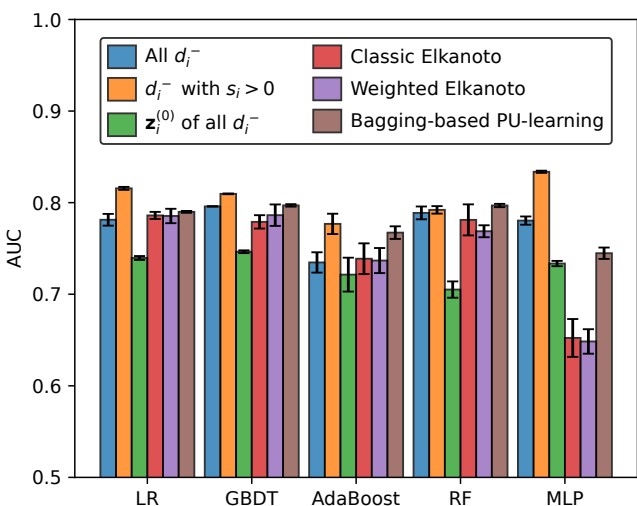

Figure 11: AKI prediction performance of widely adopted classifiers for three different settings of our proposal and three PU learning baselines.

### G.3 COMPARISON WITH POSITIVE-UNLABELLED (PU) LEARNING METHODS

While our primary focus centers on effective cohort discovery among negative samples rather than rectifying the asymmetry between positive and negative examples for enhanced performance, our approach holds relevant implications for the latter. Therefore, we compare our approach against three PU learning methods: Classic Elkanoto (Elkan & Noto, 2008), Weighted Elkanoto (Elkan & Noto, 2008), and Bagging-based PU-learning (Mordelet & Vert, 2014). Specifically, we train these

three baseline methods, treating negative samples as unlabeled data, and evaluate their performance on the testing data.

The experimental results of these baseline methods in terms of AUC (mean $\pm$ std) from five repeats are presented in Figure 11. Among the benchmarked baselines, Bagging-based PU-learning outperforms the other two methods and also surpasses the performance of the "all $d_i^-$" setting, where all negative/unlabeled samples are included in the training. This validates the effectiveness of Bagging-based PU-learning, achieved through its bootstrap aggregating techniques. On the other hand, both Classic Elkanoto and Weighted Elkanoto fail to achieve satisfactory performance. They merely marginally outperform the "all $d_i^-$" setting when employing LR and AdaBoost. This observation suggests that the "selected completely at random" assumption inherent in these two baselines may not hold in our hospital-acquired AKI prediction utilizing real-world EMR data.

In contrast to these baselines, the "$d_i^-$ with $s_i > 0$" setting of our proposal, which filters out the negative samples with negative data Shapley values, consistently achieves substantially higher AUC values across different classifiers. This firmly establishes the superiority of our approach in identifying negative samples in real-world medical data, which further underscores the validity of our constructed Negative Sample Shapley Field, thus providing a robust foundation for subsequent cohort discovery.

### G.4 COMPARISON WITH INFLUENCE FUNCTION-BASED DATA VALUATION

As an alternative data valuation technique, influence functions (Weisberg & Cook, 1982) measure how the prediction model changes when the weight of a single sample is altered slightly. In this experiment, we compare our proposed data valuation technique, specifically data Shapley values for negative samples, against influence functions for negative samples. To perform this comparison, we employ the standard leave-one-out (LOO) method (Ghorbani & Zou, 2019; Rozemberczki et al., 2022) to calculate the influence function value for each negative sample.

The cohort discovery results of influence function-based data valuation for AKI prediction are presented in Figure 12. Figure 12(a) displays the histogram of influence function values among all negative samples, which can be fitted by a single Gaussian distribution. To focus our analysis on the most influential negative samples, we set a threshold of 60% and exclude the lower 60% of negative samples based on their influence function values. Figure 12(b) illustrates the distribution of all negative samples in terms of their influence function values in the manifold space. The extracted 40% of samples with high influence function values (points brighter than dark blue) are scattered throughout the space and are not well-separated from the samples with low values (points in dark blue). Subsequently, we apply DBSCAN to the 40% negative samples with high influence function values, but it fails to generate meaningful clustering results, resulting in only one single cluster (Figure 12(c)). These cohort discovery results indicate that influence function-based data valuation does not reveal meaningful medical cohorts for AKI prediction.

Further, we compare the AKI prediction performance achieved by a new setting, "$d_i^-$ with $IF > 0$," which includes only the negative samples with positive influence function values, with three different settings of our proposal. The experimental results in terms of AUC (mean $\pm$ std) from five repeats are shown in Figure 13. It is observed that influence functions underperform data Shapley values ("$d_i^-$ with $IF > 0$" vs. "$d_i^-$ with $s_i > 0$") and generally achieve lower AUC than using all negative samples (the setting "All $d_i^-$"), indicating that the setting "$d_i^-$ with $IF > 0$" deletes informative data samples from all negative samples rather than retaining them. Consequently, influence function-based data valuation is not a suitable technique for AKI prediction.

It is worth noting that data Shapley values and influence functions have also been compared as data valuation techniques in prior work (Ghorbani & Zou, 2019), where data Shapley values exhibit a significant performance advantage over influence functions. This aligns with our findings from the evaluation results, confirming the superiority of data Shapley values over influence functions. Moreover, influence functions, which are computed as the performance difference of the prediction model with and without a specific negative sample, do not satisfy equitability conditions. This limitation of influence functions arises from their inability to account for a sample's complex interactions with other samples (Ghorbani & Zou, 2019). However, this limitation deviates from our focus on cohort discovery, because the influence of a single sample, as measured by influence functions, is not expected to be high if any cohorts exist in our context of cohort discovery. In this sense, influence

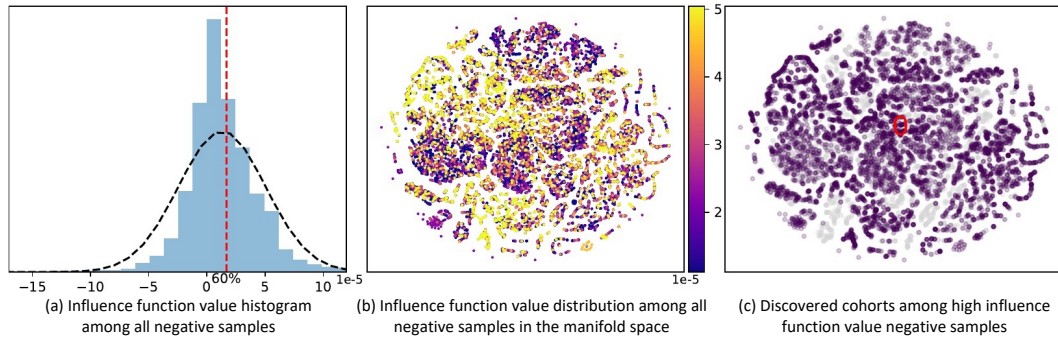

(a) Influence function value histogram among all negative samples

(b) Influence function value distribution among all negative samples in the manifold space

(c) Discovered cohorts among high influence function value negative samples

Figure 12: Cohort discovery of influence function-based data valuation for AKI prediction.

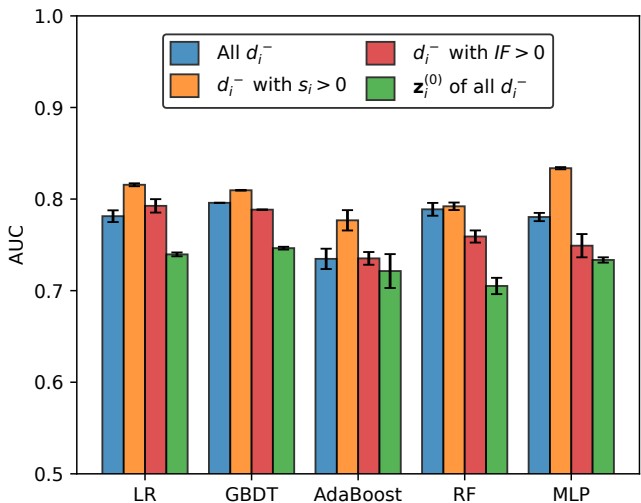

Figure 13: AKI prediction performance of widely adopted classifiers for three different settings of our proposal and influence function-based data valuation.

functions may be more suitable for identifying rare or isolated medical cases, which, although interesting, falls outside the scope of our research. Additionally, influence functions have also been identified to have robustness issues in prior work (Ghorbani et al., 2019).

## G.5 COMPARISON WITH $p$-BASED DATA VALUATION STRATEGY

In this comparison experiment, we employ a $p$-based data valuation strategy, where $p$ represents the model's predicted probability of a negative sample belonging to the negative class, i.e., the model's confidence in the negative sample's label. The cohort discovery results using this $p$-based data valuation strategy are presented in Figure 14. Figure 14(a) shows the histogram of $p$ values among all negative samples, which can be fitted by a beta distribution with the corresponding parameters $\alpha$ and $\beta$. We apply a threshold of 60% to extract the top 40% negative samples with high $p$ values. Figure 14(b) illustrates the distribution of all negative samples in terms of their $p$ values in the manifold space, with the extracted 40% high-$p$ samples (brighter than dark blue) scattering throughout the space and not being separated from the low-$p$ samples in dark blue. We then adopt DBSCAN to the 40% high-$p$ samples, resulting in unsatisfactory clustering outcomes with only two clusters as shown in Figure 14(c). This demonstrates that $p$-based data valuation is not suitable for AKI prediction.

Additionally, we devise a setting, "$p > 0.5$," which incorporates only the negative samples for which the prediction model is confident. We compare the performance of this newly introduced setting with three different settings of our proposed approach using AUC values (mean $\pm$ std from five repeats) in Figure 15. It is evident that "$p > 0.5$" performs less effectively than our proposed "$d_i^-$ with

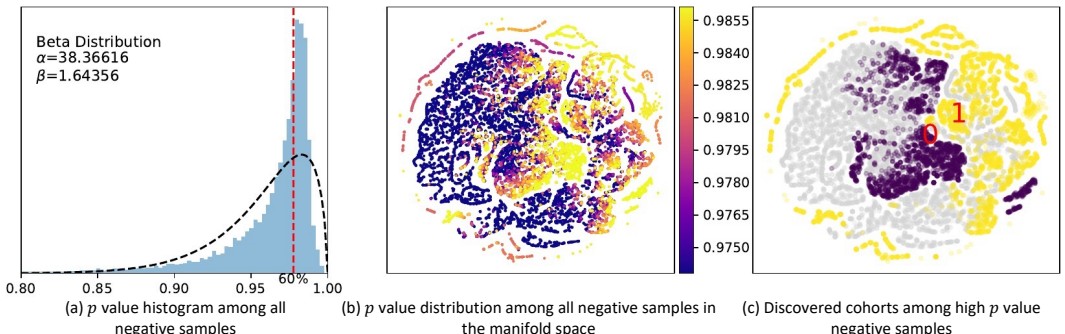

(a) $p$ value histogram among all negative samples

(b) $p$ value distribution among all negative samples in the manifold space

(c) Discovered cohorts among high $p$ value negative samples

Figure 14: Cohort discovery of $p$-based data valuation for AKI prediction.

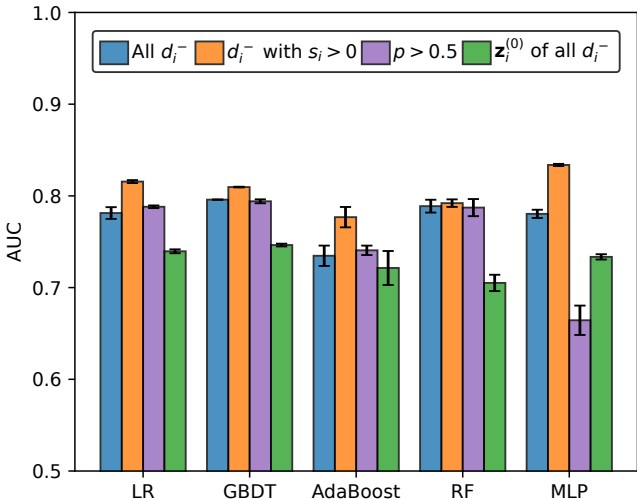

Figure 15: AKI prediction performance of widely adopted classifiers for three different settings of our proposal and $p$-based data valuation.

$s_i > 0$" approach and even exhibits worse performance than including all negative samples ("All $d_i^-$") for three classifiers (GBDT, RF, and MLP). This is because by excluding the samples with low $p$ values in the "$p > 0.5$" setting, we eliminate samples that the model is not sufficiently confident about. Consequently, this exacerbates the overfitting issue, which is more pronounced in the MLP classifier, leading to a significant drop in AUC for "$p > 0.5$" as shown in Figure 15. In summary, this $p$ value does not serve as an appropriate data valuation strategy in our AKI prediction scenario.

### G.6    COMPARISON WITH DEEP CLUSTERING BASELINES

In recent years, the concept of deep clustering, which involves simultaneously optimizing representation learning and clustering, has gained increasing attention. We further compare our proposed cohort discovery approach with two deep clustering baselines: the deep clustering network (DCN) (Yang et al., 2017) and the deep embedded K-means clustering (DEKM) (Guo et al., 2021). Specifically, DCN achieves joint dimensionality reduction by training a deep neural network alongside K-means clustering, while DEKM transforms the embedding space further to a new space to reveal cluster-structure information.

The comparison of cohort discovery results among high data Shapley value negative samples for our approach, DCN, and DEKM is shown in Figure 16. Both deep clustering baselines primarily identify one large cohort and a few smaller cohorts (indicated in the legend by the number of samples per cohort). Notably, these results resemble those obtained using cPCA in Figure 10 and our proposal without the isotropy constraint in Figure 9(b) in the ablation study. A shared characteristic of these methods is their two-step process: (i) dimensionality reduction of raw data using methods such as

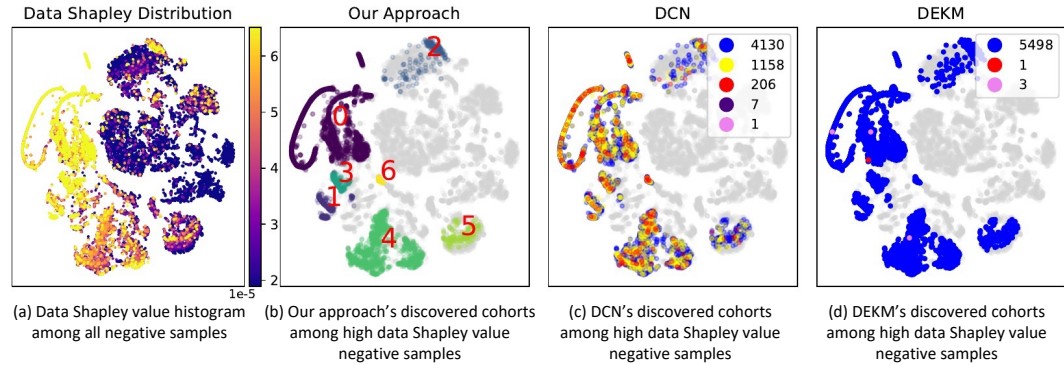

(a) Data Shapley value histogram among all negative samples

(b) Our approach's discovered cohorts among high data Shapley value negative samples

(c) DCN's discovered cohorts among high data Shapley value negative samples

(d) DEKM's discovered cohorts among high data Shapley value negative samples

Figure 16: Comparison of cohort discovery results between our approach, DCN and DEKM.

AE, SAE, SDAE, or cPCA, and (ii) clustering on the embedded representations using K-means or DBSCAN.

However, without imposing our proposed isotropy constraint to ensure uniform changes in data Shapley values across orientations, these methods fail to identify medically meaningful cohorts. This is on account of our key insight that in clinical practice, patients directed to our hospital's division of nephrology without an AKI diagnosis may represent different types with distinct symptoms, implying diverse cohorts with varied data Shapley value distributions. Only by integrating the data Shapley value-based constraint into representation learning and imposing this assumption can we effectively discover medically relevant cohorts. This Shapley-based exploration of interrelationships between samples is our primary innovation for real-world clinical practice. In summary, these experimental results further demonstrate the effectiveness of our proposed cohort discovery approach, underscoring its medical significance.

### G.7 COMPARISON WITH CLUSTERING ALL NEGATIVE SAMPLES

We conduct a comparison between our approach's cohort discovery results among high data Shapley value negative samples and results among all negative samples. The experiment results are displayed in Figure 17, with Figure 17(a) illustrating the data Shapley value distribution among all negative samples in the manifold space, and Figures 17(b) and 17(c) serving as counterparts for comparison. In Figure 17(b), grey points represent samples either possessing low data Shapley values or being labeled as noise by DBSCAN. Conversely, in Figure 17(c), grey points exclusively correspond to samples labeled as noise by DBSCAN.

Our approach, even after incorporating all negative samples for clustering, remains capable of identifying representative cohorts primarily composed of high data Shapley value negative samples. There are only a few instances of merging with the large cluster of low data Shapley value negative samples.

It is essential to highlight that negative samples with high data Shapley values significantly contribute to the prediction task. They exhibit symptoms leading to hospital admissions and are valuable from a medical perspective. In contrast, negative samples with low data Shapley values, whether showing negative values or being close to zero, are not the primary focus of our cohort discovery investigation. They may either hinder the prediction task or represent complex, healthy patients, possibly introducing noise or errors into the model.

### G.8 EXPERIMENTAL RESULTS ON THE MIMIC-III DATASET

We evaluate our proposed cohort discovery approach on the widely recognized MIMIC-III dataset (Johnson et al., 2016). This dataset is esteemed as a benchmark in healthcare analytics and comprises EMR data for dozens of thousands of patients who are admitted to intensive care units (ICU) between 2001 and 2012. Our objective is to predict in-hospital mortality on the MIMIC-III dataset, employing laboratory test data as our input. We define each patient admission as an individual sample if the duration of the admission exceeds 48 hours. Subsequently, we label each admission

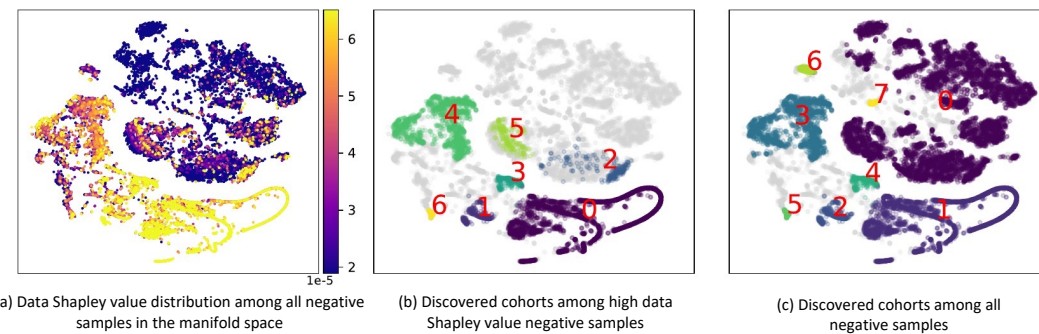

(a) Data Shapley value distribution among all negative samples in the manifold space

(b) Discovered cohorts among high data Shapley value negative samples

(c) Discovered cohorts among all negative samples

Figure 17: Comparison between clustering high data Shapley value negative samples and clustering all negative samples.

Table 3: Key statistics of MIMIC-III dataset for in-hospital mortality prediction

| Statistics | MIMIC-III Dataset |
|---|---|
| # of admissions | 51826 |
| # of positive samples | 4280 |
| # of negative samples | 47546 |
| # of lab tests | 428 |
| Input Window | 48 hours |

by assessing whether the patient passes away during their stay at the hospital. We utilize a set of 428 laboratory tests within a 48-hour "Input Window" to predict the likelihood of mortality for each admission. Prior to using these test values as input features, we apply min-max standardization, followed by averaging the results. The key statistics pertaining to the MIMIC-III dataset for in-hospital mortality prediction are summarized in Table 3.

To validate the broad applicability of our proposed approach, we conduct the same cohort discovery analysis on the MIMIC-III dataset as described, with the results shown in Figure 18. By fitting the data to a Gaussian mixture model, we unveil three distinct components within the data Shapley value histogram, as illustrated in Figure 18(a). To focus our investigation on the third component that holds high value for the prediction task, we set a 50% threshold to exclude the lower 50% negative samples based on their data Shapley values while retaining the remaining 50% for further analysis. Figure 18(b) displays the data Shapley value distribution of all negative samples in the manifold space. Subsequently, we perform DBSCAN on the extracted 50% of negative samples with high data Shapley values, successfully identifying fifteen distinct cohorts, as presented in the t-SNE plots in Figure 18(c). These fifteen cohorts effectively partition the third component of Figure 18(a) into respective Gaussian distributions, as demonstrated in Figure 19, confirming consistent data Shapley values within each identified cohort. This affirms the robustness and capability of our cohort discovery approach in decomposing samples with high data Shapley values into medically relevant and distinct cohorts.

These cohort discovery results hold promise in enhancing our comprehension of ICU patients and are poised to make valuable contributions to patient care and survival through systematic and cross-sectoral analytical research. Our experimental findings underscore the efficacy of our proposed approach on this established public benchmark dataset, highlighting its potential for broader application in other medical datasets, thus showcasing its versatility and robustness.

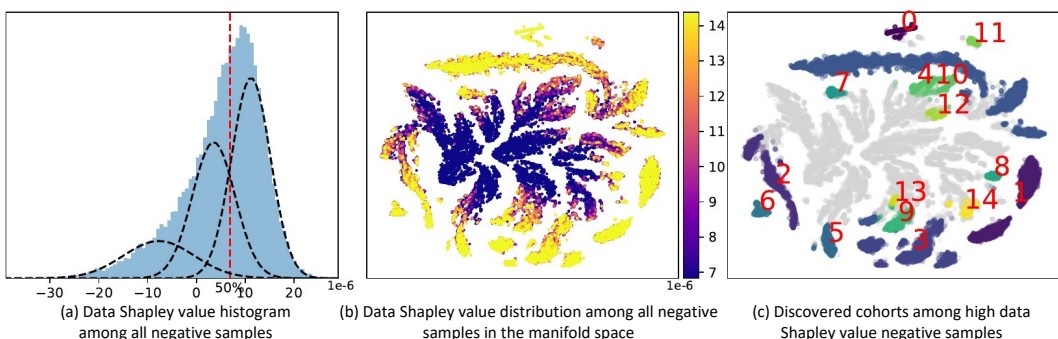

(a) Data Shapley value histogram among all negative samples

(b) Data Shapley value distribution among all negative samples in the manifold space

(c) Discovered cohorts among high data Shapley value negative samples

Figure 18: Cohort discovery of our proposal for in-hospital mortality prediction on the MIMIC-III dataset.

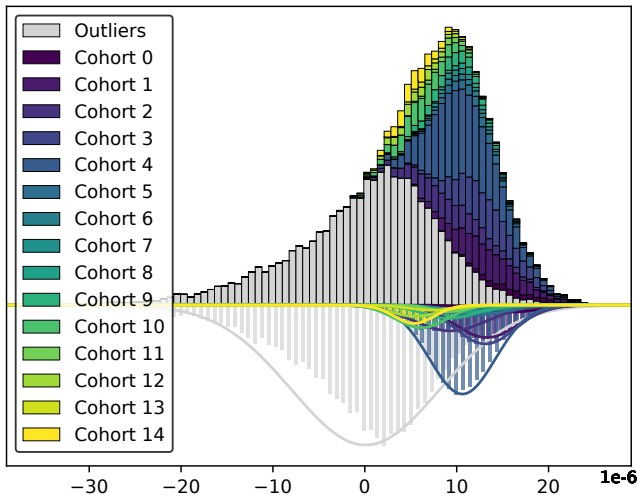

Figure 19: Data Shapley value histogram of the samples within our discovered cohorts on the MIMIC-III dataset.

