# OpenReview forum: "Exploiting Negative Samples: A Catalyst for Cohort Discovery in Healthcare Analytics"
_ICLR.cc/2024/Conference — Submitted to ICLR 2024_

### Official Review · Reviewer_73KN · 2023-10-29

**Soundness:** 3 good
**Presentation:** 4 excellent
**Contribution:** 3 good
**Rating:** 6
**Confidence:** 4

**Summary:**

The authors present a novel clustering framework motivated by strong clinical evidence that samples with negative labels are defined in an open-ended manner to highlight their potential to further understand the clinical outcome of interest. To achieve this, the authors i) use the data Shapley value to focus on a "representative" subset of negative samples and ii) enforce representation learning to preserve the interrelationship in data samples with respect to the computed Shapley value for clustering. The authors evaluate the proposed framework on a single private medical dataset with an extensive qualitative analysis to support the discovered clusters with domain knowledge. However, the evaluation is done only on a single private dataset and lacks a comparison with other clustering methods, including both traditional and cutting-edge methods. This limitation hinders the understanding of its broader impact.

**Strengths:**

1.	The paper is well-written.
2.	The authors present a novel clustering concept motivated by strong clinical evidence. This work focuses on investigating heterogeneity in the negative cohort, which can further improve the understanding of the clinical outcome of interest compared to focusing on the overall or positive cohort.
3.	The authors provide extensive qualitative analysis of the discovered clusters with strong supporting evidence.

**Weaknesses:**

1.	Although the authors provided an extensive qualitative analysis of the clusters discovered by the proposed method, it is limited to a single private dataset, which hinders its broader impact to other medical datasets.
2.	The key idea of the proposed work is conducting clustering on a subset of negative samples in the manifold that preserves meaningful structure about the data Shapley values. Considering such contribution, comparing it to other conventional and cutting-edge clustering methods (e.g., DCN [A], MOE-VAE [B]) will improve the clinical utility of the proposed work.
3.	The authors should include clustering results when clustering is performed on i) including all negative samples and ii) including positive samples to substantiate the claim of the importance of focusing on the subset of negative samples with high Shapley values.

[A] Yang et al., “Towards K-means-friendly Spaces: Simultaneous Deep Learning and Clustering,” ICML 2017.

[B] Kopf et al., “Mixture-of-Experts Variational Autoencoder for clustering and generating from similarity-based representations on single-cell data,” PLOS Comp. Biology, 2021.

**Questions:**

1.	As shown in Figure 9 (in Appendix G), the discovered clusters can be different depending on $P_{min}$, leading to different scientific discovery. How can we choose $P_{min}$ for a given dataset?
2.	There are different types of clustering methods. Why did the author choose DBSCAN over other clustering methods, such as k-means or its variants? What would be the impact of using other clustering methods?

---

> ### Author Response · Authors · 2023-11-19
> **Response to Weakness 1**
>
> **Weakness 1**
>
> We have extended the evaluation of our proposed cohort discovery approach to the widely recognized MIMIC-III dataset [c] in Appendix G.8. The MIMIC-III dataset serves as a prominent benchmark in healthcare analytics, collecting EMR data from dozens of thousands of patients admitted to intensive care units (ICU) between 2001 and 2012. In this particular evaluation, our focus centers on predicting in-hospital mortality, leveraging laboratory test data as input. Specifically, we consider each patient admission, representing an individual visit to the hospital, as a sample if its duration exceeds 48 hours. The mortality label for each admission is determined by whether the patient passes away during the specific hospital admission or not. Utilizing a set of 428 laboratory tests, we predict the likelihood of mortality for each admission.  A summary of the key statistics pertaining to the MIMIC-III dataset for in-hospital mortality prediction is provided in Table 3.
>
> To validate the broad applicability of our proposed approach, we have conducted the same cohort discovery analysis on the MIMIC-III dataset as described, with the results shown in Figure 18. In particular, through the application of a Gaussian mixture model, we identify three distinct components within the data Shapley value histogram, as depicted in Figure 18(a). To focus our analysis on the third component, which holds significant relevance to the prediction task, we adopt a 50\% threshold to exclude the lower 50\% of negative samples based on their data Shapley values, while retaining the remaining 50\%. The data Shapley value distribution of all negative samples in the manifold space is visualized in Figure 18(b). Subsequently, by performing DBSCAN on the extracted 50\% of negative samples exhibiting high data Shapley values, our approach successfully discerns fifteen distinct cohorts, as demonstrated in the t-SNE plots in Figure 18(c). These fifteen cohorts effectively decompose the third component of Figure 18(a) into respective Gaussian distributions, as confirmed in Figure 19, validating consistent data Shapley values within each identified cohort. Compared to the results found in the AKI prediction, we can see that in terms of ICU mortality prediction, the negative sample cohorts are more distinct and prominent in the constructed negative sample Shapley field. This aligns with expectations, as the negative samples in mortality (i.e., cases where patients are successfully resuscitated and discharged alive) among patients on the brink of death admitted to ICU often stem from causes that are more multi-sourced and more explicit, compared to those in a single-specialty nephrology department. This substantiates the capability of our cohort discovery approach in partitioning samples with high data Shapley values into medically relevant and distinct cohorts.
>
> The outcomes of this cohort discovery evaluation hold substantial promise in advancing our comprehension of ICU patients, and they are poised to contribute significantly to patient care and survival, particularly when combined with systematic and cross-sectoral analytical research. Our experimental results underscore the effectiveness of our proposed approach on this well-established public benchmark dataset, demonstrating its potential for broader application in the analysis of other medical datasets, thus highlighting its versatility and robustness. Further details regarding the evaluation of our approach on the MIMIC-III dataset can be found in Appendix G.8.
>
> [c] Johnson, Alistair EW, et al. "MIMIC-III, a freely accessible critical care database." Scientific data 3.1 (2016): 1-9.

---

> ### Author Response · Authors · 2023-11-19
> **Response to Weakness 2**
>
> **Weakness 2**
>
> We have investigated the two recommended deep clustering methods: the deep clustering network (DCN) [A] and the mixture-of-experts similarity variational autoencoder (MoE-Sim-VAE) [B]. DCN achieves joint dimensionality reduction through a deep neural network and K-means clustering, and we have integrated it as a clustering baseline for comparison. MoE-Sim-VAE employs a variational autoencoder (VAE) with a decoder consisting of a mixture-of-experts (MoE) architecture. However, due to its high application specificity and numerous parameters for clustering organ-specific single-cell RNA-seq data as in its officially released implementation, we have not included it in our comparison but will discuss it in the related work section.
>
> For a comprehensive evaluation, we have also adopted another cutting-edge deep clustering method recommended by [d], the deep embedded K-means clustering (DEKM), which transforms the embedding space further to a new space to reveal cluster-structure information.
>
> The comparison between our proposed cohort discovery approach, DCN, and DEKM is presented in Appendix G.6, with cohort discovery results shown in Figure 16. In terms of results, neither of the newly introduced baseline methods can identify significant cohorts as effectively as our proposed approach does. Instead, they classify the majority of points into one large cluster. These results align with those obtained using cPCA in Figure 10 and our proposal without the isotropy constraint in Figure 9(b) in the ablation study. A common characteristic of these methods lies in first a dimensionality reduction of raw data using methods, e.g., AE, SAE, SDAE, or cPCA, and a subsequent clustering on the embedded representations using either K-means or DBSCAN. This indicates that the subtle differences resulting from specific implementations of dimensionality reduction and clustering methods do not lead to significant differences in cohort discovery in the application scenario.
>
> The key difference between them and our proposed approach lies in the absence of our proposed isotropy constraint, which ensures uniform changes in data Shapley values across orientations, facilitating the identification of high-temperature areas in the field. These results substantiate the innovative contribution we have made in this paper by incorporating data valuation into medical cohort discovery. This highlights the importance of our approach, as traditional methods solely based on feature similarity struggle to effectively differentiate between different cohorts in complex and high-dimensional medical applications. This Shapley-based exploration of interrelationships between samples is our primary innovation for real-world clinical practice.
>
> In summary, the newly supplemented comparison results with DCN and DEKM, along with the previous results of cPCA and our proposal without the isotropy constraint, further validate the effectiveness of our proposed cohort discovery approach, underscoring its medical value.
>
> Further, we would like to emphasize that for spatial clustering, we employ DBSCAN on account of its well-acknowledged superior performance and its capability to discover arbitrarily shaped clusters (Ester et al., 1996; Gan \& Tao, 2015; Schubert et al., 2017). It is essential to clarify that the primary contribution of our proposal lies in exploring negative samples for cohort discovery through a Shapley-based exploration of interrelationships between these negative samples. Consequently, after constructing the negative sample Shapley field, the choice of the subsequent spatial clustering method is somewhat orthogonal to our primary contribution. Therefore, we remain open to the possibility of replacing DBSCAN with more innovative and better-performing spatial clustering alternatives, should they become available, to further enhance the cohort discovery results for the benefit of clinicians.
>
> [d] https://paperswithcode.com/task/deep-clustering
>
> [e] Guo, Wengang, Kaiyan Lin, and Wei Ye. "Deep embedded K-means clustering." 2021 International Conference on Data Mining Workshops (ICDMW). IEEE, 2021.

---

> ### Author Response · Authors · 2023-11-19
> **Response to Weakness 3**
>
> **Weakness 3**
>
> We wish to reiterate that our primary objective in this work is to discover cohorts within the negative samples that contribute significantly to the prediction task. This focus allows us to facilitate clinicians' understanding of the disease from a population perspective through the insights derived from these valuable negative samples.
>
> In response to the first suggested experiment (i), we have included a comparison between our approach's cohort discovery results among high data Shapley value negative samples and results among all negative samples in Appendix G.7. The experimental results are presented in Figure 17. Our approach, even after incorporating all negative samples for clustering, continues to identify representative cohorts composed of high data Shapley value negative samples, with only a few merging with the large cluster of low data Shapley value negative samples. It is important to emphasize that cohorts with high positive data Shapley values demonstrate a clear ability to facilitate the model to identify positive samples, indicating their fundamental value in medical research, and hence, we only focus on them in this paper. In contrast, the medical significance of cohorts without high positive data Shapley values (negative or near zero) becomes very unclear. They can either hinder the prediction task by potentially being erroneous or noisy data points, as indicated by negative data Shapley values, or they may represent complex and healthy samples of limited relevance, with data Shapley values close to zero. Consequently, these negative samples are not the primary focus of our cohort discovery efforts.
>
> Regarding the second suggested experiment (ii), where both positive and negative samples participate in the clustering process, it is crucial to understand that our proposal, based on the Shapley-based exploration of interrelationships between negative samples, relies on the premise that valuable cohorts among negative samples should exhibit similar distributions with high data Shapley values. This premise assumes a consistent label, i.e., negative samples. When labels are mixed, including both negative and positive samples, this assertion no longer holds. Specifically, the interpretation of positive samples within our constructed negative sample Shapley field becomes unclear. Therefore, such positive samples cannot be incorporated into our approach.

---

> ### Author Response · Authors · 2023-11-19
> **Response to Questions 1 and 2**
>
> **Question 1**
>
> In our assessment of clustering quality for a given dataset, we rely on the Silhouette score as a metric. The Silhouette score measures both the cohesion and separation of clusters, providing a robust evaluation of clustering quality for each parameter setting. In the case of our AKI dataset, we conduct a comprehensive evaluation and determine that setting $P_{min}$ to 100 yields the most suitable parameter choice for our DBSCAN clustering, based on the Silhouette score assessment. Further details regarding the selection of parameters for the DBSCAN algorithm are elaborated in Appendix E.3.
>
> In addition, Figure 9 in Appendix G.1 showcases how we vary the settings of $P_{min}$ to highlight the advantages of our proposal, over the variant approach without the isotropy constraint more clearly and thoroughly. As shown in Figure 9, the results of our proposal, which includes the isotropy constraint, remain relatively stable across varying $P_{min}$ settings. In contrast, when the isotropy constraint is removed, effective cohort discovery through DBSCAN is hindered, regardless of the chosen $P_{min}$ values. These results underscore the pivotal role of the introduced isotropy constraint in ensuring robust cohort discovery outcomes that are resilient to variations in spatial clustering algorithm parameters. Ultimately, this constraint contributes to the identification of medically meaningful cohorts.
>
> It is important to note that the cohort discovery results should be viewed as preliminary outcomes for subsequent complex medical research and investigation. Clinicians will delve into the identified cohorts for in-depth analysis on a population level. Therefore, the selection of parameter settings should align with the requirements of the ensuing medical research, including considerations such as available resources and manpower, particularly with regard to how many patient groups (out of the patients of interest) need to be partitioned and analyzed for detailed investigation.
>
> **Question 2**
>
> In our previous response to Weakness 2, we have already addressed this issue to some extent. Here, we briefly reiterate the key points for clarity.
>
> Firstly, we regard DBSCAN as a mature and proven spatial clustering tool with broad applicability across various scenarios. Unlike K-means, DBSCAN does not require a pre-assumed number of clusters (K) and is capable of identifying high-density connected subspaces of arbitrary shape.
>
> In contrast, as also noted in the reference paper [A], K-means is generally suited for clustering data samples that are evenly distributed around their centroids in the feature space. This assumption may not hold in our context of high-dimensional medical data. Furthermore, our approach seeks to identify high-temperature connected subspace in the constructed negative sample Shapley field, aligning well with DBSCAN's mechanism.
>
> Secondly, we emphasize that our approach is not confined to a specific spatial clustering method. Indeed, the particular implementation of clustering is orthogonal to our overall cohort discovery framework. Our work is open to any other clustering methods, and the corresponding substitution is straightforward.
>
> Finally, we stress that the ultimate aim of our work is to assist in identifying medically meaningful cohorts in clinical cases, especially in negative samples, to enhance our understanding of diseases and benefit clinical practice. In the current setup, our approach already provides insightful and valuable contributions to this end.

---

> > ### Comment · Reviewer_73KN · 2023-11-20
> > **Response to the rebuttal**
> >
> > I thank the authors for the rebuttal. I found the response to be thorough and appropriate. I've adjusted my score as such. I believe it would greatly benefit readers to include qualitative analysis on MIMIC-III to provide valuable insights for future use.

---

### Official Review · Reviewer_T3Vm · 2023-10-30

**Soundness:** 3 good
**Presentation:** 3 good
**Contribution:** 3 good
**Rating:** 6
**Confidence:** 4

**Summary:**

In this work, the authors introduce the "negative sample Shapley field" to represent the distribution of negative samples in clinical predictive tasks. This field is derived using a stacked denoising autoencoder and Shapley values. Experiments were conducted on a real-world AKI dataset, and the authors explored the potential clinical significance of the identified clusters.

**Strengths:**

The exploration of negative samples as proposed is both novel and clinically relevant. The methodology is clearly presented, and the paper is generally well-structured.

**Weaknesses:**

Please see the questions below.

**Questions:**

My primary comments are:

1. The authors claim that negative samples with zero Shapley values typically indicate healthy individuals without any overt AKI-related symptoms. Does a high Shapley value correspond to a high likelihood of AKI? If so, why not use the predicted probability of each sample directly?

2. How are negative samples defined? Do they refer to ground-truth negative samples or predicted negative samples? How are false negatives/positives addressed? Is there a chance that one of the identified clusters pertains to FN/FP cases?

3. How is the clinical significance of these cohorts determined in section 4.2?

4. How are the predictions formulated? Are raw features, or are the learned embeddings used as inputs for LR?

5. What are the broader implications of the proposed method and its conclusions? Can they offer insights for future modeling or labeling processes in ML/DL research?

6. How is the DAE stacked? Are the reconstruction loss and iso loss applied to each layer of the stacked DAE?

---

> ### Author Response · Authors · 2023-11-19
> **Response to Questions 1 and 2**
>
> **Question 1**
>
> The data Shapley value of a negative sample serves as a measure of its contribution to the prediction task. Therefore, focusing on the second component in the middle of Figure 2(a), we observe that its data Shapley values are centered around a mean value close to zero. This implies that these particular negative samples provide minimal contributions to the prediction task, suggesting that they are generally healthy without overt symptoms related to AKI. It is crucial to emphasize that the data Shapley value of a negative sample is distinct from the model's predicted probability, i.e., the likelihood of AKI. Specifically, the former quantifies the individual contribution of each negative sample, while the latter characterizes the relationship between each negative sample and the label.
>
> Our primary objective lies in taking into account data valuation when conducting spatial clustering analysis among negative samples, and we propose to assess each negative sample based on its data Shapley value. This approach enables a Shapley-based exploration of interrelationships between the samples. To offer a comprehensive comparison, we have introduced a $p$-based data valuation strategy, where $p$ represents the model's predicted probability of a negative sample being classified into the negative class, indicating the model's confidence in the assigned label. A detailed comparison between this baseline and our proposed approach is provided in Appendix G.5.
>
> The cohort discovery results using the $p$-based data valuation strategy are presented in Figure 14. Figure 14(a) displays the histogram of $p$ values among all negative samples, which fits well with a beta distribution. Applying a threshold of 60\%, we extract the top 40\% high-$p$ negative samples. Figure 14(b) illustrates the distribution of all negative samples in terms of their $p$ values in the manifold space. The top 40\% high-$p$ samples appear scattered throughout the space and are not clearly separated from the low-$p$ samples in dark blue. Subsequent application of DBSCAN to the 40\% high-$p$ samples results in unsatisfactory clustering outcomes, forming only two clusters, as depicted in Figure 14(c).
>
> Additionally, we introduce a setting "$p>0.5$," which incorporates only the negative samples for which the prediction model has high confidence. We compare the performance of this newly introduced setting with our proposed approach's three different settings in terms of AUC. The results, shown in Figure 15, clearly demonstrate that "$p>0.5$" performs worse than our proposal "$d_i^-$ with $s_i>0$" and even underperforms the inclusion of all negative samples ("All $d_i^-$") for GBDT, RF, and MLP classifiers. This inferior performance is attributed to the removal of samples with low $p$ values in the "$p>0.5$" setting, resulting in the exclusion of samples that the model is not sufficiently confident about. Consequently, this exacerbates the overfitting issue, which is particularly pronounced in the MLP classifier, as evidenced by the significant degradation in AUC for the "$p>0.5$" setting shown in Figure 15.
>
> Based on these two experiments comparing the $p$-based data valuation strategy and our data Shapley values for negative samples, it is evident that the $p$ value is not an appropriate data valuation strategy in the context of AKI prediction.
>
> At the same time, we refer you to our response to Weakness 3(ii) of Reviewer KsRk's reviews, where we also discuss a similar concern, explaining the difference between data valuation and model confidence (predicted probability).
>
> **Question 2**
>
> We would like to clarify that "negative samples" are cases where patients visit our nephrology department but are not diagnosed with AKI. Therefore, both the positive and negative samples in our paper are ground-truth labels, based on actual clinical diagnosis results. This work does not involve any predicted labels, and therefore, does not involve any considerations or concerns related to FN/FP.

---

> ### Author Response · Authors · 2023-11-19
> **Response to Questions 3, 4, 5 and 6**
>
> **Question 3**
>
> The cohort discovery within negative samples facilitated by our proposed approach has yielded representative cohorts, the medical patterns of which are thoroughly validated by our clinicians, as demonstrated in Section 4.2. The clinical significance of these discoveries is profound. The cohorts identified among negative samples have the potential to unveil future positive cases, pathological correlations, or conditions that bear similarities. This reciprocal relationship between negative and positive samples has the potential to redefine and refine the concept of positive samples in theoretical medical research.
>
> We would like to underscore that, as front-line clinicians and medical researchers who have focused on AKI for many years, the samples, whether positive or negative, are not merely data points within a dataset. They represent real-world patients who we have been assisting and investigating in the context of AKI in clinical practice. Consequently, the medical insights derived from our proposed cohort discovery approach deepen our clinicians' understanding of the disease, AKI, from a population perspective. This understanding has the potential to bring about substantial benefits in terms of patient management, medical resource allocation, and the advancement of medical research in the long term.
>
> **Question 4**
>
> The raw features, consisting of 709 lab tests, serve as inputs for LR to calculate the data Shapley value for each negative sample. The detailed procedure is presented in Algorithm 1 of Appendix C.2.
>
> We note that in the subsequent phase of manifold learning through SDAE, these calculated data Shapley values for negative samples play a crucial role in the computation of the isotropy constraint loss term, as described in Equation (6). This constraint ensures uniform changes in data Shapley values across all orientations. Hence, our proposed manifold learning with structure preservation and isotropy constraint renders the constructed negative sample Shapley field more suitable for subsequent cohort discovery, as exemplified in the transition from Figure 1(b) to Figure 1(c).
>
> **Question 5**
>
> The novelty of our proposed cohort discovery approach stems from its innovative extension of the data Shapley value concept to enable a Shapley-based exploration of interrelationships between samples. This extension moves beyond traditional feature-based similarity methods, asserting that valuable cohorts should exhibit similar distributions with high data Shapley values.
>
> Let us illustrate with a practical example: when a patient exhibits symptoms of upper respiratory distress, outpatient physicians typically prioritize measuring the patient's body temperature and inquiring about any recent travel history or epidemiological contacts. This essentially constitutes leveraging an identified negative sample cohort for efficient medical delivery, specifically for the diagnosis of high-risk infectious viral diseases such as COVID. This is crucial because, in this scenario with limited medical resources, it is impractical to triage all coughing patients into isolation wards.
>
> In this routine example, the "patient without fever" can be considered to exhibit a high data Shapley value in the context of the COVID task, representing a high-temperature zone within the proposed negative Shapley field. Although our proposed approach stems from clinical experience and the examination of relevant medical data, we believe that this innovative approach offers valuable insights for other application domains (e.g., to discover typical classification and exclusion strategies in complex problems). Its recognition aids in deepening the understanding of negative samples, particularly in discerning the typical patterns of negative samples specific to certain tasks.
>
> **Question 6**
>
> In the process of stacking multiple DAEs, we adhere to the standard practice introduced in the original SDAE paper by Vincent et al. (2010). The model architecture of SDAE-based manifold learning is presented in Figure 6 of Appendix D.1, where both the reconstruction loss and the isotropy loss are applied to each individual DAE of SDAE.
>
> To provide a more detailed view, Figure 6(a) illustrates the computational details of the $k$-th DAE, where both the reconstruction loss and the isotropy loss are employed in this process. Both losses collectively contribute to the training of robust feature extractors, leveraging clean and corrupted inputs simultaneously. After training the encoder of the $k$-th DAE, it is applied to clean inputs for transformation, as depicted in Figure 6 (b). This iterative training process continues, as shown in Figure 6 (c), until the last DAE generates the final encoded representation within the manifold space. This encoded representation serves as the foundation for subsequent cohort discovery.

---

> ### Comment · Reviewer_T3Vm · 2023-11-20
>
> Thanks to the authors for their comprehensive replies. I would stand for my score.

---

### Official Review · Reviewer_KsRk · 2023-10-30

**Soundness:** 3 good
**Presentation:** 3 good
**Contribution:** 3 good
**Rating:** 6
**Confidence:** 3

**Summary:**

The authors explore the concept that in some domains, here medicine, negative samples are simply those that are "not known to be positive". In these cases, the negative-ness of a label is uncertain. The authors use a Shapley based strategy for evaluating the negatives. An autoencoder is used to project the negative samples into a lower dimensional space, in which clustering is performed via DBSCAN. Finally, clusters with extreme Shapley values are offered as cases that deserve review.

**Strengths:**

The authors highlight a subtlety of labeling that is often overlooked. In this case, the "noisy label" concept is a poor fit, since the negative labels may be inappropriate not-at-random. The approach is then demonstrated on a challenging real-world problem.
Overall, the approach is reasonable and the analysis is section 4.3 helps to validate the importance of each element.

The value of the uncovered cohorts seems to be in (i) their self-similarity, by which I mean that members within a cohort are similar and (ii) the (presumed) causal relationship between the factors that define a cohort with the outcome of interest. Clustering achieves the first of these aims and Shapley analysis achieves the second, albeit not by identifying or understanding causality in any way but rather only by analysis of correlation.

**Weaknesses:**

1. While the proposed approach is reasonable, the authors do not offer any alternatives or baselines. One that immediately comes to mind is influence-functions, which can offer much the same information as Shapley values for this application.

2. The insights generated by the proposed approach are not dissimilar to what one might see using feature-importance analysis of other types. Can these be compared with the proposed method?

3. Of what value are the identified cohorts? If some are valuable because they are high risk or similar to positives (negative Shapley values), then isn't this equivalent to proximity to the decision boundary? How does this compare to looking at model confidence? Having identified a cohort of patients that might require special treatment, can they be reliably identified in the future? Does the fact that the authors using clustering guarantee that cohorts are separable (not necessarily linearly)?

**Questions:**

See weakness 3.

---

> ### Author Response · Authors · 2023-11-19
> **Response to Weakness 1-1**
>
> **Weakness 1**
>
> We would like to clarify that previously, we compared our proposed approach against several baseline methods. These baselines included contrastive principal component analysis and positive-unlabelled learning methods, and the corresponding results were discussed in Appendices G.2 and G.3, respectively.
>
> The central concept of our proposed approach revolves around incorporating data valuation into spatial clustering analysis among negative samples, through a Shapley-based exploration of interrelationships between the samples. We choose to utilize data Shapley values for data valuation in this process, given that data Shapley values are recognized as a popular and well-established equitable data valuation technique in recent years, offering distinct advantages over other data valuation techniques examined in related work (Ghorbani \& Zou, 2019; Rozemberczki et al., 2022).
>
> We appreciate the reviewer's suggestion of including influence functions [a] in our comparison. Influence functions also serve as a data valuation technique, which assesses how the prediction model changes when the weight of a single sample is slightly altered. To provide a more comprehensive experimental evaluation, we have incorporated influence functions as a baseline in our study.
>
> Specifically, in line with standard practice for comparing with influence functions, we employ the leave-one-out (LOO) method (Ghorbani \& Zou, 2019; Rozemberczki et al., 2022). We would like to note that the comparison between data Shapley values and influence functions in terms of data valuation is also evaluated in Ghorbani \& Zou (2019), demonstrating a significant performance advantage for data Shapley values over influence functions.
>
> Our evaluation results for influence functions are detailed in Appendix G.4. Specifically, we present the cohort discovery results of influence function-based data valuation for AKI prediction in Figure 12. Figure 12(a) displays the histogram of influence function values among all negative samples.
> The distribution appears as a single normal distribution with a mean of zero, which is different from the distribution of data Shapley presented in Figure 2(a).
> Figure 12(b) illustrates the distribution of all negative samples in terms of their influence function values in the manifold space.
> From this figure, we cannot discern significant high-temperature areas, as samples with different temperatures are completely mixed together.
> This suggests that if influence functions are used for data valuation, they do not exhibit a clear trend of spatial proximity similarity in the feature vector space, which may make it challenging to effectively benefit spatial clustering algorithms for further cohort discovery.
> This concern is further experimentally validated, as shown in Figure 12(c). We implement the same subsequent clustering steps as in Figure 2(c) in the negative sample space based on influence functions (i.e., selecting 40\% of high influence function values as input for DBSCAN).
> From Figure 12(c), it can be seen that DBSCAN fails to generate meaningful clustering results and only produces a single cluster.
> These cohort discovery results indicate that influence function-based data valuation does not reveal meaningful medical cohorts for AKI prediction.
>
> Next, we compare the AKI prediction performance achieved by a new setting, "$d_i^-$ with $IF>0$," which includes only the negative samples with positive influence function values, with three different settings of our proposal. Figure 13 presents the experimental results in terms of AUC (mean $\pm$ std) from five repeats. It is observed that influence functions underperform data Shapley values, i.e., "$d_i^-$ with $IF>0$" vs. "$d_i^-$ with $s_i>0$", and generally achieve lower AUC than using all negative samples (the setting "All $d_i^-$"). This indicates that the setting ``$d_i^-$ with $IF>0$'' deletes informative data samples from all negative samples rather than retaining them.
> Consequently, this set of controlled experiments confirms that, compared to influence functions, data Shapley values are a more suitable and effective data valuation measure for the discovery of the relevant medical cohort in AKI that we focus on.

---

> ### Author Response · Authors · 2023-11-19
> **Response to Weakness 1-2**
>
> Furthermore, it is essential to highlight that influence functions, calculated as the performance difference of the prediction model with and without a particular (negative) sample, do not satisfy equitability conditions. This limitation arises due to their failure to account for complex interactions with other samples (Ghorbani \& Zou, 2019). However, this limitation does not align with our research focus on cohort discovery.
> Specifically, if there is a specific cohort, meaning there is a cluster of samples with a sufficient number of samples in it, and they are close to each other in the feature space and in terms of their data valuation, then the influence function value of any single sample in that cohort should be relatively low.
> Hence, influence functions may be more suited for identifying rare or isolated medical cases, which, while interesting, falls outside the scope of our research. Additionally, prior work has identified robustness issues associated with influence functions [b].
>
> In summary, we have provided additional comparison results between our approach and influence functions, addressing the reviewer's suggestion. We did not include these results previously due to their relatively inferior performance and their deviation from our primary research focus on cohort discovery. It is worth noting that the comparison between data Shapley values and influence functions has also been explored in prior studies (Ghorbani \& Zou, 2019), as mentioned above. Our supplementary experimental results are consistent with those reported in the existing literature.
> Based on our experimental findings,
> data Shapley values are validated as a more suitable data valuation technique in our specific application scenario compared to influence functions.
>
> [a] Cook, R. D. and Weisberg, S. Residuals and influence in regression. New York: Chapman and Hall, 1982.
>
> [b] Ghorbani, Amirata, Abubakar Abid, and James Zou. "Interpretation of neural networks is fragile." Proceedings of the AAAI conference on artificial intelligence. Vol. 33. No. 01. 2019.

---

> ### Author Response · Authors · 2023-11-19
> **Response to Weakness 2**
>
> **Weakness 2**
>
> Regarding this comment that mentions "the insights generated by the proposed approach," we think that these insights should pertain to the findings presented in Figure 3 of the submitted paper. We would like to elucidate that the lab test patterns showcased in Figure 3 offer a comparative view between the cohort of interest against all the other samples outside the cohort, in terms of the lab test value probability density among the involved samples. Through this analysis, we aim to unveil the distinctive characteristics of each discovered cohort, to further investigate its clinical significance, which has been validated by clinicians. It is essential to emphasize that the lab test patterns displayed in Figure 3 pertain to samples within a specific cohort compared to all the other samples. Consequently, various dominant features emerge within different cohorts.
>
> Feature-importance analysis, on the other hand, typically measures the importance of each feature in predicting the output of the model, from our understanding. In other words, the feature-importance analysis can obtain the distribution of features based on labels. However, for our application scenario, the process is reversed, meaning we obtain cohorts based on the similarity of feature distributions by constructing a negative sample Shapley field. From this perspective, these two types of work are difficult to compare with each other because they are tools designed to address different problems. We note that in the medical context which we are focusing on, cohorts do not have objective labels. We hope that the explanation above clarifies your concern, and if you still expect specific comparative experiments, we would greatly appreciate more specific details on this matter.

---

> ### Author Response · Authors · 2023-11-19
> **Response to Weakness 3-1**
>
> **Weakness 3**
>
> Please let us address each of the points raised in this comment:
>
> **(i) Of what value are the identified cohorts?**
>
> The identified cohorts among negative samples hold significant medical value as they unveil medically meaningful patterns, as exemplified in Section 4.2 of our paper. These patterns shed light on the investigated medical problem, in this case, AKI prediction. Further, the discovered cohorts can reveal potential future positives, pathological correlations, or similar conditions. This reciprocal relationship between negative and positive samples can contribute to defining positive samples in theoretical medical research. Such insights can significantly enhance clinicians' comprehension of the disease and the distribution of visiting patients during practical clinical consultations.
>
> Let us illustrate with a more specific example. In this paper, we propose a cohort discovery approach on AKI negative sample cases admitted to our hospital's nephrology department, resulting in the identification of several cohorts (discussed in Section 4.2). Among them, our further investigation leads us to classify Cohort 2 as an "inflammatory cohort."
>
> We believe this cohort reveals a clinical insight: patients with infections are often treated with higher doses of antibiotics, significantly increasing the metabolic burden on the kidneys. In some instances, this heightened renal metabolic load can manifest as renal symptoms, leading to the patients being triaged to nephrology for diagnosis and treatment.
>
> However, even in cases of severe infection, physicians adhere to standard dosages of antibiotics, meaning that, in most cases, this does not lead to further renal damage (AKI). From a data perspective, these cases form a cohort with distinctive infection characteristics and antibiotic usage, representing a cohort among negative samples.
>
> The clinical insights derived from the analysis of this cohort are of great medical value. For instance, nephrologists, when assessing patients, consider the toxicity of antibiotics administered recently. If a patient's profile matches these characteristics, a more conservative approach and further observation may be preferred. This is because the renal symptoms in such patients are likely not rooted in kidney damage.
>
> Traditionally, clinical treatment strategies based on cohort insights rely on human experience, and considering the complexity of clinical treatment, these experiences often tend to be one-sided. In the era of data science, we can use learning techniques to help us gain a deeper understanding of diseases and their treatments from new perspectives.

---

> ### Author Response · Authors · 2023-11-19
> **Response to Weakness 3-2**
>
> **(ii) If some are valuable because they are high risk or similar to positives (negative Shapley values), then isn't this equivalent to proximity to the decision boundary? How does this compare to looking at model confidence?**
>
> First and foremost, we would like to clarify that our research is focused on identifying and understanding patient cohorts, which represent a kind of statistical knowledge. The fundamental goal of the proposed approach is to assist us in understanding the pattern of typical negative cases encountered in clinical practice, rather than focusing on individual cases or any particular patient.
>
> We would like to emphasize that the data Shapley value for a negative sample quantifies the negative sample's contribution to the prediction model's performance, which is a distinct concept from the model's predicted probability of a negative sample belonging to the negative class or the model's confidence in the negative sample's label, denoted as $p$. To provide a more comprehensive evaluation, we have adopted a $p$-based data valuation strategy and compared it against our proposed approach. Detailed experimental results are presented in Appendix G.5. Based on the cohort discovery results of the $p$-based data valuation strategy in Figure 14 and the comparison in AKI prediction performance in Figure 15, it is evident that $p$-based data valuation is not suitable for AKI prediction. For a more in-depth discussion about the differences between these two data valuation strategies, please refer to our response to Question 1 in Reviewer T3Vm's reviews and Appendix G.5.
>
> We further clarify that as described in Section 4.1, we focus our investigation on the negative samples, specifically those with positive data Shapley values, as they are the ones contributing positively and are particularly valuable for the prediction task. Although data valuation may not have a direct relationship with the confidence of a model (or, in other words, the decision boundary), a negative data Shapley value implies that its inclusion as a training sample in the training set has a detrimental effect on the overall accuracy of the model. This may indicate that the sample is likely far from the so-called decision boundary (on the side of incorrect classification). Because samples with negative data Shapley values do mislead the model in terms of performance, the model may align more confidently with incorrect classifications.
> Our focus on negative samples with positive data Shapley values is because these patients typically exhibit certain symptoms that warrant their admissions to our hospital's division of nephrology, rather than random hospital visits. Therefore, such samples tend to be of high research value from a medical perspective and deserve further exploration. However, negative samples with negative data Shapley values, which are detrimental to the prediction task, can be complex cases, which is not the scope we prioritize exploring.
>
> In summary, data valuation measures the contribution of a sample to the performance of the final trained model when used in the training set. This measurement is not directly related to the model's confidence in classifying a specific sample (i.e., its decision boundary).
>
> In the existing literature, the data Shapley value that we have adopted, as well as the influence function you mentioned earlier, are considered different techniques within the realm of data valuation. However, it is rare to see $p$ values used in data valuation.
>
> From another perspective, $p$ values tend to indicate the relationship between a specific model and a sample, rather than elucidating the representativeness of a data sample for the prediction problem itself. Recall that the computation of the data Shapley value is based on subsets formed from all permutations of training samples and involves the evaluation of numerous models trained on these subsets. The same applies to the influence function. In contrast, the $p$ value is based on measurements from a single, particular model.

---

> ### Author Response · Authors · 2023-11-19
> **Response to Weakness 3-3**
>
> **(iii) Having identified a cohort of patients that might require special treatment, can they be reliably identified in the future?**
>
> We think that the example mentioned earlier regarding the positive impact on clinical practice from negative sample cohort identification addresses this question to some extent.
>
> We acknowledge that our proposed approach for cohort discovery is still in its early stages and cannot replace clinicians with their medical expertise. Instead, it plays a supportive role in real-world clinical practice. Although we have conducted quantitative comparisons with several baselines, our current cohort discovery results are primarily qualitative. We provide our clinicians with discovered medical patterns per cohort, facilitating their understanding of the studied disease (AKI) from a population perspective rather than an individual one. This can benefit patient management and medical resource allocation in the long run. The validation by our clinicians, in turn, underscores the substantial medical value and utility of our approach.
>
> **(iv) Does the fact that the authors using clustering guarantee that cohorts are separable (not necessarily linearly)?**
>
> In real-world scenarios, cohorts may overlap as a patient's medical conditions can be highly complex, spanning multiple cohorts. However, in our adopted density-based spatial clustering technique DBSCAN, the identified cohorts do not overlap and are separable. As mentioned earlier (referring back to our response to (iii)), our primary aim is to enhance clinicians' understanding of patients visiting the hospital regarding AKI on a population level rather than providing insights on an individual level. This approach is valuable in the context of cohort discovery.

---

> ### Comment · Reviewer_KsRk · 2023-11-22
> **Author responses**
>
> I would like to thank the authors for their very comprehensive responses. I appreciate the addition of more baselines and the MIMIC-III dataset. However, I still struggle on two points. First, I don't think it's appropriate to simply mention that baseline comparisons are in Appendix G, without comment on what the baselines are what the general findings of the comparisons are or other details. The proposed concepts are not so wholly novel that consideration of alternative approaches is deserving of relegation to the appendix. Second, the concept explored by the authors is fundamentally focused on negative labels, yet claims of value are focused on cohorts. Why analyze negatives if not specifically to understand their properties in relation to positives? Otherwise, one might as well analyze all patients regardless of label. This gets somewhat lost in the narrative.
>
> All together, I think the paper has improved but not enough to jump from 6 to 8. I will stand by my score.

---

> > ### Author Response · Authors · 2023-11-22
> > **Re: Author responses-1**
> >
> > We appreciate your valuable feedback regarding our rebuttal. In addressing the two new concerns raised, we would like to provide the following responses:
> >
> > ### **Positioning of Baseline Methods**
> >
> > We concur with your observation that placing the baseline in the appendix may not be the most optimal choice, as this could hinder readers from fully grasping the design philosophy and rationale of the proposed solution when only reading the main text, thereby limiting their comprehensive understanding of our proposed approach.
> >
> > In response to this, we have updated our manuscript to address this issue. Specifically, we have expanded Section 4.3 of the main text to include discussions on potential alternative approaches for each component of the proposed cohort discovery process, as well as the reasons why our approach is a more reasonable choice in the context of the problem at hand. Additionally, we have linked this to the more targeted and detailed discussions provided in the appendix.
> >
> > We believe these modifications will effectively aid the readers' comprehension. If you have additional suggestions for improvements in writing, we would also appreciate your further input.

---

> > ### Author Response · Authors · 2023-11-22
> > **Re: Author responses-2**
> >
> > ### **Innovation and Significance of Negative-Sample-Based Cohort Discovery**
> >
> > In the field of medical data analytics, it is widely recognized that discovering patient cohorts represents a highly motivated yet extremely challenging issue. Therefore, our research focuses on exploring the unique domain-specific characteristics, which are not captured by traditional/general methods, to further enhance the outcomes.
> >
> > As emphasized in the introduction of our paper, in the context of medical applications, there exists an inherent asymmetry between positive and negative samples. This asymmetry stems from the clinical diagnostic process: positive samples are confirmed based on strict definitions of specific diseases, whereas negative samples represent the complement.
> >
> > This means that for positive samples, we have a relatively comprehensive understanding of their attributes for specific diseases. In other words, due to the rigor of the diagnostic process, the determination of positive samples is based on the most stringent and adequate criteria, with key data features concentrated around the diagnosed condition. This reveals a fact in EMR scenarios: existing properties determine the positive labels. However, for negative samples, the situation is quite different, as it is challenging to articulate the data features of individuals who have not been diagnosed.
> >
> > Our research is inspired by this innovative observation: the asymmetry between positive and negative samples and the inadequate exploration of negative sample data distribution. Consequently, we believe that this under-utilization of negative samples has limited the potential of traditional patient cohort discovery methods.
> >
> > Addressing your concern, **"Why analyze negatives if not specifically to understand their properties in relation to positives?"** is fundamental to our methodological philosophy.
> >
> > The properties of diagnosed positive samples are well-defined (their clarity is the reason these cases are positive), allowing for studying all of them as a whole (and in-depth research has already been conducted for most specific diseases). Therefore, in our current scenario, further cohort analysis of positive samples is unnecessary or not a primary focus.
> >
> > However, understanding why certain individuals do not have a disease is a complex question in clinical practice. Some cases might present for inexplicable reasons, while others could be due to our limited understanding of diseases or conservative diagnostic processes. Thus, discovering cohorts among negative samples helps us find clues in these complex situations to better understand the actual conditions of the cases, which is beneficial for clinical practice. Driven by this rationale, we introduce the concept of data valuation into the further investigation of negative samples.
> >
> > Regarding another point you raised, **"one might as well analyze all patients regardless of label".**, we can intuitively understand it as follows: in cases where positive samples have a high data valuation, it indicates that they align well with the "medical standards for diagnosing specific diseases." However, since medical standards already exist, such results may not yield more intriguing conclusions.
> >
> > Conversely, for the multitude of complex negative samples, we believe that data valuation can provide crucial information, especially for high-value cases. This implies that we have substantial evidence that these cases positively contribute to the model's recognition of positive samples, creating a significant contrast with the "positives defined by medical standards." When clear clusters with similar features are discernible from a data perspective, it largely indicates a patient group worth studying as a whole, aligning with the medical definition of a cohort. This process represents a new discovery of knowledge, whose practical medical significance we have already explained to you in the form of examples in our previous responses.
> >
> > In summary, our proposal, grounded in clinical practice and recognizing the uniqueness of the field, emphasizes the asymmetric nature of positive and negative samples and advocates for a more comprehensive exploration of negative samples. This approach, under-considered by traditional methods and previous studies, has proven by our results to be effective in advancing the challenging task of patient cohort discovery. This validates the title of our paper, "Exploiting Negative Samples: A Catalyst for Cohort Discovery in Healthcare Analytics."

---

### Author Response · Authors · 2023-11-19
**Global Response**

We would like to extend our profound gratitude to all the reviewers for their insightful comments and constructive suggestions, which have played a pivotal role in elevating the quality of our paper. Below, we summarize our comprehensive responses to the reviewers' comments.

### **Distinction Between Data Shapley Values and $p$ Values**
We have emphasized that the data Shapley value of a negative sample is distinct from the model's predicted probability, i.e., the likelihood of AKI, denoted as $p$. Specifically, the former quantifies the individual contribution of each negative sample, while the latter characterizes the relationship between each negative sample and the label. These two concepts, serving different purposes, are neither analogous nor interchangeable.  This clarification is expanded upon in our responses to Weakness 3 of Reviewer KsRk's reviews and Question 1 of Reviewer T3Vm's reviews.

### **Clinical Significance of Our Work**
We have underscored the medical significance of our proposed approach, particularly in identifying cohorts within negative samples that may indicate future positive cases, pathological correlations, or conditions that bear similarities. Furthermore, this reciprocal relationship between negative and positive samples has the potential to redefine and refine the concept of positive samples in theoretical medical research. Further details are provided in our responses to Weakness 3 of Reviewer KsRk's reviews and Questions 3 and 5 of Reviewer T3Vm's reviews.

### **Addition of New Experiments**
In alignment with reviewers' suggestions, we have integrated five new experiments into our study. These include:
* a comparison with influence function-based data valuation in Appendix G.4, in response to Weakness 1 of Reviewer KsRk's reviews;
* a comparison with $p$-based data valuation strategy in Appendix G.5, in response to Weakness 3 of Reviewer KsRk's reviews and Question 1 of Reviewer T3Vm's reviews;
* a comparison with deep clustering baselines in Appendix G.6, in response to Weakness 2 of Reviewer 73KN's reviews;
* a comparison with clustering all negative samples in Appendix G.7, in response to Weakness 3 of Reviewer 73KN's reviews;
* experimental results of our approach on the MIMIC-III dataset in Appendix G.8, in response to Weakness 1 of Reviewer 73KN's reviews.

### **Individual Responses to Reviewer Comments**
We have meticulously addressed each comment from the reviewers in separate, detailed responses below.

We believe that the aforementioned revisions have substantively enhanced the quality and rigor of our paper, which we hope will meet the high standards of the reviewers and contribute meaningfully to the field of machine learning for healthcare. If there are any additional questions or comments, we stand ready and eager to engage in further discussions.

---

### Meta-Review · Area_Chair_FiKn · 2023-12-13

**Metareview:**

This paper has been assessed by three knowledgeable reviewers who unanimously yet independently scored it as marginally acceptable. The authors have engaged the reviewers in a discussion and provided detailed responses to raised questions. These responses were very helpful indeed, but they only caused one reviewer to upgrade their score. The reviewers have highlighted the need to reflect multiple alternative analytic approaches that can be feasibly considered for the stated task. They have also suggested specific approaches to comparisons, and suggested benchmarking against a well-established data repository such as MIMIC. The proposed approach is intriguing and even though the foundational novelty of it is limited, it could have impact on practical applications. Despite that in its current form the paper is below the ICLR acceptance threshold, the authors are strongly encouraged to incorporate the constructive feedback received and produce a much more robust version of it for the next submission deadline. That should not be too hard to accomplish.

**Justification For Why Not Higher Score:**

This paper is almost there, but there is a number of obvious improvements that can be done to make it much stronger.

**Justification For Why Not Lower Score:**

n/a

---

### Decision · Program_Chairs · 2024-01-16

Reject